# African cratonic lithosphere carved by mantle plumes

Nicolas Luca Celli [1,2]*, Sergei Lebedev [1], Andrew J. Schaeffer[3] & Carmen Gaina [4]

How cratons, the ancient cores of continents, evolved since their formation over 2.5 Ga ago is debated. Seismic tomography can map the thick lithosphere of cratons, but its resolution is low in sparsely sampled continents. Here we show, using waveform tomography with a large, newly available dataset, that cratonic lithosphere beneath Africa is more complex and fragmented than seen previously. Most known diamondiferous kimberlites, indicative of thick lithosphere at the time of eruption, are where the lithosphere is thin today, implying surprisingly widespread lithospheric erosion over the last 200 Ma. Large igneous provinces, attributed to deep-mantle plumes, were emplaced near all lithosphere-loss locations, concurrently with or preceding the loss. This suggests that the cratonic roots foundered once modified by mantle plumes. Our results imply that the total volume of cratonic lithosphere has decreased since its Archean formation, with the fate of each craton depending on its movements relative to plumes.

[1] Dublin Institute for Advanced Studies, Dublin, Ireland. [2] Trinity College Dublin, Dublin, Ireland. [3] Geological Survey of Canada, Pacific Division, Sidney Subdivision, Natural Resources Canada, Sydney, BC, Canada. [4] Centre for Earth Evolution and Dynamics (CEED), University of Oslo, Oslo, Norway. *email: nlscelli@gmail.com

The mantle roots of cratons are thought to have been coupled to the overlying crust since their Archean formation and stabilization[1]. Cratonic mantle lithosphere is compositionally buoyant, thick (over 200 km), cold and mechanically strong, which is probably what enabled the cratons to survive intact for over 2.5 Ga[2]. The occasional destruction of the cratonic mantle lithosphere is well documented, but its mechanisms are debated. It has been attributed to the effects of fluids and metasomatism caused by subduction or rifting and extension adjacent to the cratons[3–9]. It has also been suggested that cratonic lithosphere can be destroyed by interaction with thermo-chemical plumes rising from the deep mantle[10–13], and recent tomographic models are consistent with the thinning of cratons by plumes in parts of the Baltic Shield, Greenland and the Siberian Craton[14,15]. It remains unclear, however, if a significant proportion of the original volume of the cratonic lithosphere (as it was at, say, the Archean-Proterozoic boundary) may have been eroded or if, instead, the great majority of cratons are preserved.

Seismic tomography detects present-day cratonic lithosphere by anomalously high seismic velocities at and around 100–200 km depths (Supplementary Fig. 1), with these anomalies mainly due to the anomalously low temperatures within the thick lithosphere. Diamondiferous kimberlites and lamproites yield evidence for the existence of the characteristically thick cratonic lithosphere at the time of their emplacement[16]. Taken together, the evidence from tomography and kimberlites can offer insights into the temporal evolution of the cratonic lithosphere[17,18]. This requires, however, tomography with resolution at the relevant, regional tectonic scales and sufficiently large kimberlite databases.

The African continent is composed primarily of Precambrian terranes, assembled in the Late Neoproterozoic-Early Paleozoic Pan-African orogeny[19,20]. Three major cratons identified in Africa are the West African, Congo and Kalahari Cratons (Fig. 1a), with the smaller Tanzanian Craton located east of Congo[20]. A number of Large Igneous Provinces (LIPs)—large-scale volumes of both intrusive and extrusive igneous rocks[21]—were emplaced in Africa over the last 200 million years (My). The Central Atlantic Magmatic Province[22] (CAMP) at 200 Ma and Paraná-Etendeka[23] at 135 Ma accompanied or preceded the opening of the central and southern Atlantic Ocean, respectively. The emplacement of the Karroo LIP[24] at 180 Ma pre-dated the onset of seafloor spreading between Africa and Antarctica at 170 Ma[25]. More recently, abundant volcanism has accompanied the development of the East African Rift System (EARS) (30 Ma to present[26]).

Most tomographic models of Africa show broad high-velocity anomalies beneath its three major cratons (e.g. refs. [20,27], Supplementary Fig. 1). Until recently, however, seismic data coverage in much of Africa has been sparse, limiting the tomographic resolution.

Here, we assemble now available broadband seismic data from new stations in different parts of Africa, which significantly improve the data sampling, and combine them with a very large global dataset. We use waveform inversion to extract structural information from surface and regional S and multiple S waves recorded on the seismograms. The resulting tomographic model AF2019 shows an African lithosphere that is much more complex and fragmented than seen previously. The increased resolution of the imaging makes possible a quantitative joint analysis of the seismic and kimberlite data, revealing continual evolution of Africa's cratons over the last 200 million My.

global dataset of waveform fits of over 1.2 million vertical-component, broadband seismograms, including the newly available data from Africa[28] (Fig. 1b, Supplementary Fig. 2a). Despite the improvements, the station coverage in Africa remains uneven and relatively sparse. For this reason, it was essential to include the global data, which contained source-station pairs that sampled the Africa region only partially but yielded information complementary to that from the African stations alone. The regularisation of the model was tuned using extensive regional spike tests (Methods). The new regional data, the addition of the global data, and the area-specific regularisation resulted in a substantial improvement in resolution across Africa, compared to previously published global and regional tomographic models[20,27,29] (Supplementary Fig. 1). The model is global and contains shear-wave velocity ($V_S$) distributions beneath other continents and oceans as well; there it is similar to the published models SL2013sv and SL2013NA[29,30].

High-velocity anomalies associated with the cold cratonic lithosphere dominate the model at 100–200 km depths (Fig. 1c, Supplementary Figs. 3 and 4). Their depth extent varies from one craton to another. Underneath the EARS, a pronounced low-velocity anomaly extends from Afar in the north to Tanzania in the south and from near the surface down to the deep upper mantle and transition zone, the bottom of the model (Fig. 1c, d)[31,32]. Major low-velocity anomalies underlie the northern margins of the Red Sea and the Gulf of Aden (on the Arabian Plate, where the majority of the volcanoes are located)[33]; these anomalies extend down to 200–260 km depth (Supplementary Figs. 3 and 4).

Low-velocity anomalies are also present beneath the lithosphere of southern Africa and adjacent ocean basins to the southwest and east of it (Fig. 1d), indicating hot asthenosphere beneath the African Superswell[34]. This hot asthenosphere is likely to be responsible for at least a part of the anomalously high elevation observed.

Comparing our new model with those published previously (Supplementary Fig. 1), we observe consistency at larger scales, with all models showing pronounced high velocities beneath the three major cratons. At smaller scales, the higher resolution of our new tomography brings into focus the deep structural variations relating to regional tectonic features. For example, there is no smearing of the high-velocity anomalies, characteristic of cratonic lithosphere, into the Atlantic Ocean (which casts doubt on the notion—put forward previously and based on earlier, smoother tomographic models—that cratonic lithosphere of the Congo and other cratons extends westward beneath the Atlantic Ocean[20]). The new data available today allow us to resolve sharper boundaries of the cratonic lithosphere (Supplementary Fig. 1), located primarily within the outlines of Africa's known major cratons but with substantial complexity and fragmentation, and with a number of separate cratonic blocks outside of them (Fig. 1c). Many of the features we discuss could be seen, in a smoother form, in some of the earlier models, in particular SL2013[29]. This is to be expected: an increase in resolution adds detail, rather than changing the image entirely. The higher resolution provided by our new model AF2019 is essential in that it reveals the kimberlite-craton relationship that could not be identified using previous models. In southern Africa, for example, the Kalahari Craton in SL2013 is generally where we see it now but it is smoother and broader. The western boundary of the West African and Congo Cratons are also defined sharper in the new AF2019 than in SL2013 and other previous models.

In Fig. 2, we plot the highest-velocity (and, by inference, lowest-temperature) cores of the cratonic mantle lithosphere using 3D surfaces of positive 5% $V_S$ anomaly. This threshold

## Results

**Tomography of the African upper mantle**. Our new, upper-mantle model of Africa and surroundings is constrained by a

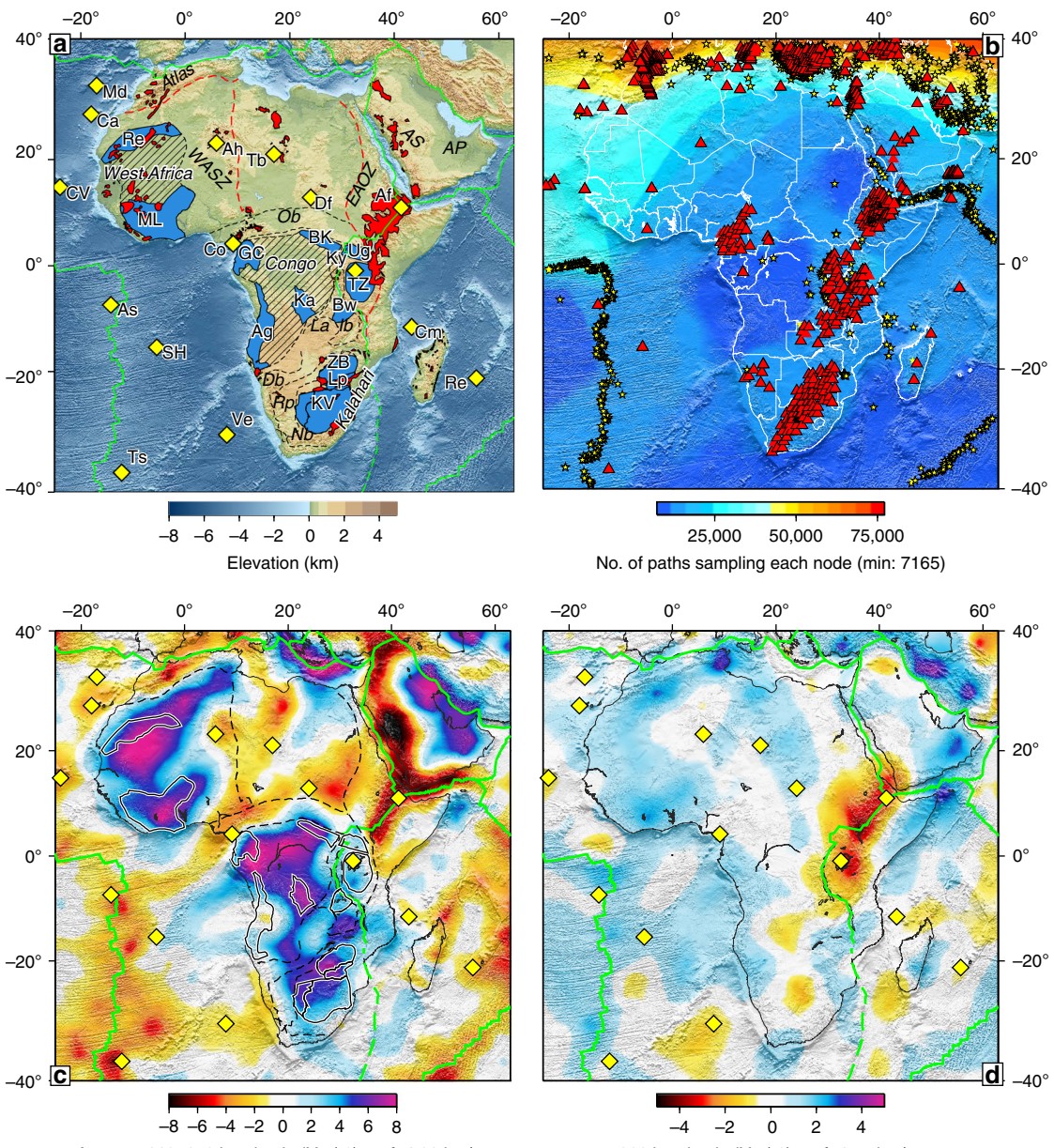

**Fig. 1 Main tectonic features, seismic data coverage and our tomography of Africa. a** Cratons and other primary features. Archean shields are plotted in blue: Re Reguibat, ML Man-Lèo, GC Gabon-Cameroon, BK Bomu-Kibali, Ug Uganda, TZ Tanzania, Ka Kasai, AG Angola, Bw Bangweulu Block, ZB Zimbabwe, Lp Limpopo Block, KV Kaapvaal. Mobile belts are plotted in black dashed lines: Ob Oubanguides, Ib Irumide, Db Damara, Nb Namaqua, La Lufilian Arc, Rp Rehoboth Province. Large Igneous Provinces and Volcanics are plotted in red, hotspots as yellow diamonds: Md Madeira, Ca Canary, As Ascension, SH Saint Helena, Ve Vema, Ts Tristan da Cunha, Cm Comoro, Re Reuniòn, Ah Ahaggar, Tb Tibesti, Df Darfur, Af Afar, Ky Kenyan. Other features: WASZ West African Shear Zone, EAOZ East African Orogenic Zone, AS Arabian Shield, AP Arabian Platform, Atlas Atlas Mountains. **b** seismic stations (red triangles) and events (yellow stars) used in tomography, plotted on the hit-count map of data sampling. **c** Average shear-wave speed ($V_S$) in the 110–150 km depth range, with geological features as in **a**. **d** $V_S$ at 330 km depth.

isolates velocities characteristic of cratonic lithosphere according to global tomography[35] and, alternatively, to temperature estimates from samples from cratonic mantle lithosphere[36] and conversion of the temperatures to seismic velocities[37]. The bottom of these cores is not the Lithosphere-Asthenosphere Boundary (LAB). However, thicker (and colder) cores do indicate where the lithosphere is the thickest (Supplementary Fig. 3), as expected from the relationship between the lithospheric thickness and temperature given by realistic geotherms[38,39]. In the thick cratonic lithosphere, the increase of temperature with depth is relatively slow and the LAB can be expected to be marked by only

a subtle change in the slope of the depth dependence of temperature and seismic velocity[40]. For this reason, direct estimates of the LAB depth from seismic tomography models are ambiguous, unless thermodynamic modelling including seismic data or models is performed[41]. For the purpose of discriminating whether or not the characteristically cold, thick cratonic lithosphere is present beneath a location, the 5% $V_S$ anomaly is an effective threshold. Our results and inferences, however, are not dependent on this particular number and also hold with a 4 or 4.5% threshold (Supplementary Note 1, Supplementary Fig. 5).

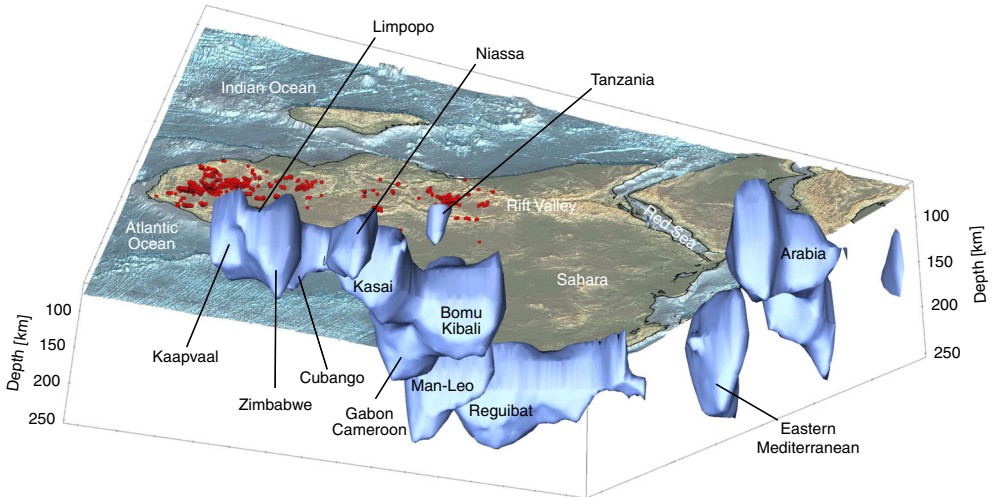

**Fig. 2 Three-dimensional representation of cratonic lithosphere in the tomographic model.** The view is looking up and to the northwest from beneath the southern Indian Ocean. The +5% $\delta V_S$ contour, plotted between 80 and 260 km depths, encloses the nuclei of the cratonic lithosphere. The bottom of the nuclei is not the lithosphere-asthenosphere boundary (that would be closer to 0% $\delta V_S$), but thicker nuclei do indicate thicker cratonic lithosphere. The Eastern Mediterranean anomaly comprises the thick, Triassic oceanic lithosphere[67] and a portion of the Hellenic subducting slab. Subduction is also seen beneath and to the north-east of Arabia. Kimberlite and lamproite locations are shown as red circles.

**Cratonic lithosphere beneath Africa.** Known Archean shields occupy only portions of the recognized African cratons (Fig. 1a). Archean basement is likely to extend beyond these shields' boundaries, but its complete extent is unknown due to the overlying sedimentary cover. Globally, there is a strong correlation between the locations of high-velocity cratonic lithosphere and the Archean crust above[35]. The presence of thick mantle lithosphere, as revealed by our tomography, thus shows where Archean crust is likely to be present as well, even if unexposed at the surface. We can also identify a number of locations with Archean crust but without cratonic mantle roots, which implies their erosion in the course of the cratons' evolution.

In West Africa, our model shows two major high-velocity bodies beneath the Man-Lèo and Reguibat Shields. Between the two lithospheric roots, the lithosphere is somewhat thinner and, thus, warmer, suggesting the existence of two, separate lithospheric units. In the westernmost parts of both the Man-Lèo and Reguibat Shields, the cratonic roots are absent.

Beneath the Congo Craton, previously identified as a single, broad, high-velocity anomaly[20,27,29] (Supplementary Fig. 1), we image three distinct, high-velocity blocks of cratonic mantle lithosphere, with different thickness beneath the Gabon-Cameroon, Bomu-Kibali and Kasai Shields[20] (Figs. 1c and 2). Between these blocks, the lithosphere is also cratonic, but thinner than within them. By contrast, the Angolan Shield in the west does not have any cratonic mantle lithosphere beneath it, apart from its northernmost tip.

Near the south-eastern boundary of the Congo Craton, we image another relatively thick, high-velocity lithospheric block, located either just within or just outside the Congo Craton, depending on the definition of the boundary[20,42]. Covered by Phanerozoic sediments, the block is characterised by higher topography compared to the neighbouring Owambo Basin to the west and the part of the Damara Belt that lies to the east of it. Diamondiferous kimberlites (Figs. 3–5), high P-wave velocities detected previously beneath the eastern part of the block[43] and gravity and heat flow data[44] provide further evidence for the presence of cratonic lithosphere beneath this unit. Because this cratonic block underlies the Cubango River basin, we identify it as the Cubango Craton. Our tomography shows that the Cubango

Craton is a few hundred km wide and forms a distinct thick-lithosphere unit within the Congo Craton (Figs. 1c, 2, 4 and 5).

In eastern Africa, a small, relatively thin high-velocity root under the Tanzania Craton is underlain by a pronounced low-velocity anomaly associated with the EARS, as seen previously in regional tomography[45]. South of Tanzania, crustal geology is complex and the definition and lithospheric age of tectonic units are debated[46–48]. Our imaging enables us to identify and map the previously unknown lateral extent of the cratonic lithosphere.

The Bangweulu Block, south-southwest of the Tanzania Craton, displays reworked Archean rocks and has been considered a craton[48,49] but is a product of Proterozoic geodynamic evolution[46]. Our model shows no cratonic root beneath this block. A recent regional seismic study[50] also shows no high velocities, and the high electrical resistivity detected near the southern boundary of the block[51] is thus likely to represent thick lithosphere of a unit to the south.

South of Bangweulu, we map, for the first time, the hidden Niassa Craton of Archean age, proposed previously to underlie some of the younger rocks of the Southern Irumide Belt based on geological data[47,48]. High seismic velocities[50] and electrical resistivities[51] have recently been detected in regional studies, but lateral extent of the anomalies remained unknown. Our results show that the thick, high-velocity lithosphere of the Niassa Craton, unexposed at the surface, extends as much as 500 km across.

The lithosphere of the Niassa Craton may have played a key role in the localisation of the deformation associated with the southward propagation of the East African Rift System[52]. Around the Tanzania Craton, the Eastern and Western Rift branches have developed along the eastern and western boundaries of this mechanically strong block. Further south, the EARS continues as the Malawi Rift, situated along the eastern boundary of the Niassa Craton (see the plate boundary in Fig. 1c). The Niassa Craton is thus likely to have determined the location of the rift and contributed to the complexity of the EARS morphology.

In southern Africa, the thickest lithosphere is beneath the western Zimbabwe Craton and Limpopo Belt, with thinner cratonic roots present beneath the northern, central and north-western Kaapvaal Craton[53]. Cratonic lithosphere is also present

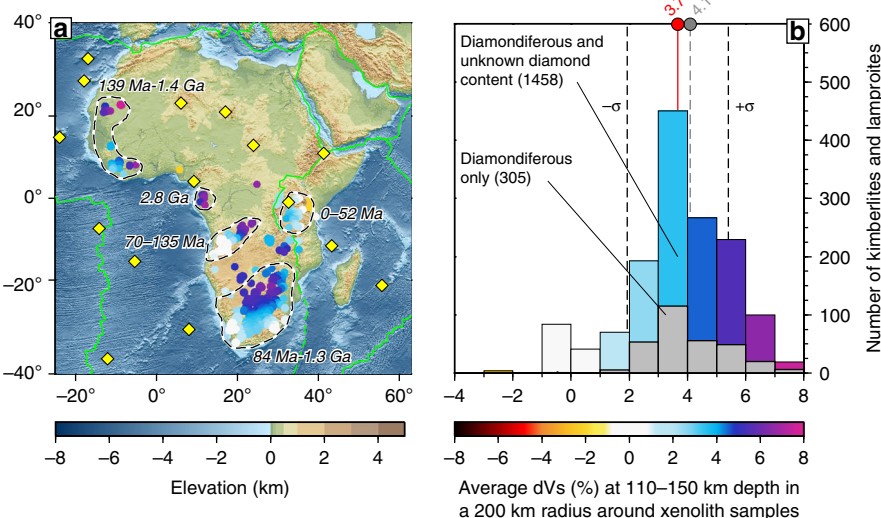

**Fig. 3 Seismic velocity in the lithosphere beneath kimberlites and lamproites.** The emplacement age ranges from[65] are indicated on the map for the different areas (dashed lines). Values of the average $\delta V_S$ over the 110–150 km depth range and within a 200-km-radius circle around each Kimberlite sample are plotted on the map (**a**) and as a histogram (**b**). $\delta V_S$ colour scale is as in Fig. 1c. The circle sizes show the lateral averaging area. Average $\delta V_S$ across all diamondiferous and unknown-diamond-content kimberlites and lamproites is shown with a red line and dot (**b**), with the standard deviation also shown. Average $\delta V_S$ for diamondiferous samples only is shown with a grey line and dot. Both groups present the same pattern: most are not on thick cratonic lithosphere at present. Elevation and bathymetry are plotted on panel **a**; plate boundaries are plotted in green.

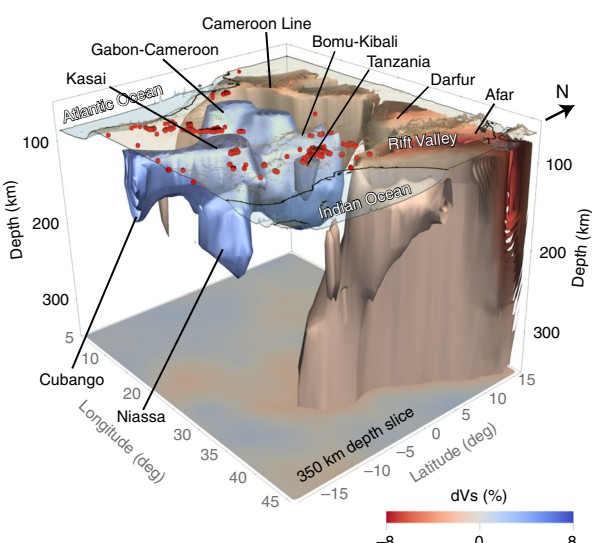

**Fig. 4 Three-dimensional view of hotspots and cratons in central-eastern Africa.** High-velocity isosurfaces at +5% and low-velocity ones at −1.5% are plotted in the 80–350 km depth range. The 350 km $\delta V_S$ map view is plotted at the depth. Kimberlite and lamproite sample locations are shown as red circles.

beneath the Paleoproterozoic Kheis-Okwa-Magondi Belt west-northwest of Kaapvaal[20], consistent with it having Archean basement[20], and beneath central and northern parts of the adjacent Rehoboth Province[54], the location of the previously proposed, unexposed Maltahohe Craton[20,55]. At shallow mantle-lithosphere depths, the Limpopo Belt stands out with relatively low $V_S$ between the Moho and ~100 km[40]. Cratonic lithosphere is notably absent beneath the southern and eastern Kaapvaal and north-eastern Zimbabwe Cratons.

The Arabia Plate separated from Africa only around 25 million years ago, with the opening of the Red Sea and the Gulf of Aden[56]. Thick cratonic lithosphere underlies the Arabian Platform in the eastern part of the plate (Fig. 1c), implying unexposed Archean crust—not known from geological data—beneath its thick sediments. The high-velocity anomaly extends north-east just across the main Zagros Thrust, but most of it is well within Arabia[57] and, therefore, is due to cratonic lithosphere rather than subduction. The south-eastern margin of the craton is just west of the Oman Mountains. The deep boundary of the Arabia Platform thus determines both the surficial boundary between the Persian Gulf and the Gulf of Oman and the structural boundary between the adjacent Zagros and Makran[57] subduction zones. Our results also indicate that the low-elevation Arabian Platform in the east and the high-elevation Arabian Shield in the west of the Arabia Plate have distinctly different lithospheres and asthenospheres (thick and thin, cool and hot, respectively), suggesting that the east-west elevation increase is primarily isostatic, in contrast with a dynamic origin postulated for it in some previous studies[58].

Given the Precambrian age of most of Africa[19,20], the complexity of its lithospheric architecture is remarkable. African cratonic lithosphere is highly fragmented, compared to the vast cratonic domains in North America, northern Eurasia or Australia (Supplementary Fig. 6). Average $V_S$ at 100–150 km depth beneath Africa is lower than the global continental average, whereas the lateral $V_S$ gradients are higher than average (Supplementary Fig. 1), indicating greater lithospheric heterogeneity and fragmentation. This points to an extensive reworking of the lithosphere. Another key observation is that a significant proportion of the Archean cratonic crust in Africa is not underlain by characteristically thick cratonic lithosphere. Beneath the western Reguibat and Man-Lèo and nearly the entire Angola Shield, and beneath southern Kaapvaal and north-eastern Zimbabwe cratons, the mantle roots are missing, which implies their erosion by mantle processes.

## Discussion

Lithospheric erosion can be mapped by comparing the present extent of cratonic lithosphere, evidenced by tomography, and its past extent, evidenced by diamond-bearing kimberlites and

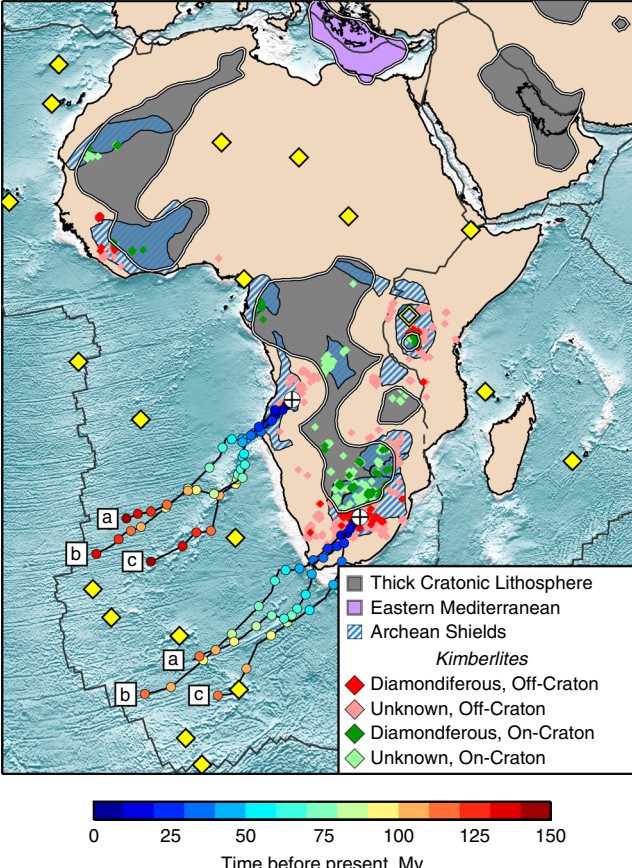

**Fig. 5 Preserved and eroded cratonic lithosphere beneath Africa and Arabia.** The thick lithosphere present today is shown in grey. The grey areas include all locations with shear-velocity anomaly exceeding +5% in the 80–150 km depth range, which indicates cold, thick cratonic lithosphere (except in eastern Mediterranean where the high velocities show a part of the Hellenic slab and the exceptionally thick, Triassic oceanic lithosphere). Geologically mapped Archean shields[20] are shown with blue and white stripes. Kimberlites and lamproites atop thick lithosphere at present are distinguished from those atop thinned lithosphere. The latter (red, pink) are indicators of lithospheric erosion. Locations of two now eroded cratons (white crossed circles) are reconstructed back in time following three different plate-tectonic reconstructions: a[73], b[72], c[67]. Present-day hotspot locations are shown as solid yellow diamonds; the southern, Tanzania end of the elongated EARS anomaly—as empty diamond.

lamproites. These rocks serve as proxies for the existence of thick cratonic lithosphere at the time of their eruption, because the pressure-temperature conditions of the diamond-stability field require the presence of thick (>150 km) lithospheric roots[16,59,60]. The minimum depth of the origin of kimberlites (~150 km) is well established, based on diamond stability and experimental petrology[16,61,62]. It is also well established that a great majority of kimberlites come from the deep lithosphere (with some, possibly, even deeper[63]) and that at these depths, temperatures that are sufficiently low to create the conditions for the diamond-stability field imply a very thick lithosphere, which occurs only in cratons[16,60,64]. Kimberlites provide estimates of both the geo-thermal gradient and composition in the lithospheric mantle at the time of their eruption and have been used extensively to map the thick Archean lithosphere beneath the cratons of Africa[20].

Comparing the distribution of 1606 kimberlite and lamproite samples in Africa[18] with $V_S$ anomalies at 110–150 km depth, we observe that at least half of the kimberlites are not in the areas of high-velocity anomalies associated with the thick lithosphere of cratons (>5%, Figs. 1–4). This is in contrast with North America, for example, where most kimberlites are on thick cratonic lithosphere[18,30] (Supplementary Fig. 6). This implies that a significant proportion of African cratonic lithosphere (including, at least, the southern Kaapvaal, western Man-Lèo, Angola and Tanzania Cratons) has been destroyed or substantially thinned since the kimberlite eruptions, during the last 200 My.

It has been suggested previously that kimberlites tend to erupt near the edges of cratonic blocks[17,18], with implications for both the origin of kimberlites and their utility as representative deep-lithosphere samples. The lithospheric craton boundaries can be mapped from our high-resolution tomography using either a threshold value or the gradient of $\delta V_S$. Regardless of how we define the boundaries, kimberlites in Africa do not demonstrate a preferential location, either closer to cratonic lithosphere boundaries or within cratonic interiors (Supplementary Fig. 5). Many other diamondiferous kimberlites and lamproites are well outside of today's thick cratonic lithosphere, indicating litho-spheric erosion.

What processes can result in the destruction of the cold, strong and compositionally buoyant roots of cratons is a matter of debate[5]. Fluids, melt infiltration and metasomatism can weaken and re-fertilize the lithospheric mantle, so that it can then be recycled into the convecting mantle. Proposed mechanisms that could promote this include subduction[6,8], rifting and stretching[9] and mantle plumes[10,13,15]. The destruction of the North China and Wyoming Cratons' lithosphere, for example, has been attributed to hydrous melts rising from subducting slabs[4,6,7]. In Africa, the diamondiferous kimberlite ages[65] require the presence of cratonic lithosphere until ~150 Ma beneath the western Man-LÃÍo Craton, ~130 Ma beneath the Angolan Shield and ~84 Ma beneath southern Kaapvaal Craton, whereas the most recent subduction in the vicinity was during the Pan-African Orogeny around 500 Ma[66], which rules out the subduction-related mechanisms.

Lithospheric stretching associated with rifting may facilitate melt infiltration and lithospheric refertilization, possibly resulting in the destruction of cratonic lithosphere. In Africa, rifting and craton-lithosphere loss happened in temporal and spatial proximity in some but not all cases. The most recent rifting episode relatively close to southern Kaapvaal Craton occurred during the ~170 Ma breakup of southern Gondwana[67], but the presence of thick cratonic lithosphere is evidenced by kimberlites until much later, up to 84 Ma[65]. More importantly, the rifting was many hundreds of kilometres away from some of the parts of southern Kaapvaal with root loss. Rifting and associated extension are thus unlikely to have caused its lithosphere's destruction.

The opening of the southern and central Atlantic Ocean was preceded by rifting at the western margins of the Congo and West African cratons. However, the fate of the cratonic lithosphere adjacent to the resultant continental margin is markedly different from one location to another. The Angolan Shield lost its cratonic lithosphere almost entirely, but the southwestern Gabon-Cameroon Shield just to the north of it has not lost any.

African cratonic lithosphere must therefore have been eroded by a different process—most likely, interaction with thermo-chemical mantle plumes[10,14,15]. LIPs, their origin commonly attributed to mantle plumes[10,22,23,26] are co-located across Africa with the cratonic lithosphere destruction that we identify. These include the Central Atlantic Magmatic Province (CAMP, 200 Ma), with magmatism on and near the West African Craton; Karroo (180 Ma) on the Kalahari Craton; Etendeka (130 Ma) on the Angola Shield; and Afar-EARS (30 Ma-present), reaching the Tanzania Craton.

We are, at present, witnessing on-going craton destruction in Tanzania, where hot asthenosphere attributed to the Kenyan

Plume[68] is in direct contact with what remains of the lithospheric keel of the craton[69] (Fig. 4). Evidence for the past existence of thick cratonic lithosphere is given by diamondiferous kimberlites aged up to 52 Ma, but younger samples (e.g. Igwisi Hills Kimberlite, 0.012 Ma[65]) are barren. This is consistent with Cenozoic erosion of the Tanzania Craton lithosphere and the loss of the pressure-temperature conditions necessary for diamond stability.

The Angolan Shield kimberlites indicate the presence of a thick cratonic root up to 124–135 Ma[65], roughly the time of the Paranã-Etendeka LIP emplacement (135 Ma[23]). At present, the Angolan Shield has no cratonic lithosphere, in stark contrast with the part of the Congo Craton just to the north, located away from the LIP and with cratonic root intact (Figs. 1 and 2). This suggests that the Angolan Shield lithosphere was eroded by the Tristan da Cunha Plume, thought to have caused the Paranà-Etendeka LIP[23].

In southern Kaapvaal, the ~180 Ma Karroo LIP[24,70] overlies the eroded part of the craton. The presence of diamondiferous kimberlites up to 84 Ma[65] indicates that cratonic lithosphere initially survived the impact of the LIP-causing plume but was, eventually, recycled into the convecting mantle ~100 My later. The lithosphere loss appears to have coincided with a pronounced uplift at the southern margin of the craton at ~80 Ma, evidenced by apatite fission track data[71]. In the western Man-Lèo Craton, the emplacement of the CAMP at ~200 Ma also preceded the loss of the cratonic keel, by at least 50 Ma.

Figure 5 shows a map of the thick cratonic lithosphere beneath Africa today, as inferred from our high-resolution tomography. It also indicates where cratonic lithosphere has been eroded—this is where diamondiferous kimberlites are not on thick lithosphere at present (red). In addition to known diamondiferous kimberlites, we also plot those with unknown-diamond content in pink (for many of these, the presence of diamonds is not listed for commercial reasons—it is well known, for example, that there are diamonds in Angola). Kimberlite ages[65] indicate that the erosion we see has occurred over the last 200 My.

For two example locations of craton-lithosphere erosion (Angola and southern Kaapvaal), we trace their movements according to Africa's absolute plate motion in recent plate-tectonic reconstructions[67,72,73] (Fig. 5). The tracks show that the cratons with missing lithosphere were, at the time of the LIP emplacement, in close proximity to the hotspots that are now beneath the South Atlantic. The differences in the tracks illustrate the uncertainty in the plate motion of Africa; we also note that the hotspots themselves are not necessarily stationary over 100 My time scales[67]. For the Angolan Shield, there is no evidence for thick cratonic lithosphere after the Etendeka LIP emplacement, when the craton was above the Tristan da Cunha Hotspot. The lithosphere of southern Kaapvaal Craton, in contrast, was lost while far from any hotspot, after 84 Ma. This confirms that, in this case, the cratonic root was recycled into the convecting mantle ~100 My after the plume impingement.

Our observations imply that the impact of a mantle plume on cratonic lithosphere can weaken and modify it sufficiently so that it is eroded and recycled into the convecting mantle, possibly enhancing and accelerating pre-existing metasomatic weakening processes[12]. The lithosphere loss can occur concurrently with or shortly after interaction with the plume (as in the case of Angola and Tanzania) or a few tens of million years later (~50 My for western Man-Lèo and ~100 My for southern Kaapvaal Cratons).

Not all cratonic lithosphere close to LIPs gets eroded. The lateral extent of the root-loss zone depends upon pre-existing lithospheric structure and how the plume interacts with it. Most root-loss areas we identify are elongated, a few hundred kilometres wide and stretching along the boundaries of the remaining cores of cratons with intact lithosphere. How plumes weaken cratonic lithosphere and how it is then removed by concurrent or

subsequent convection processes is an important outstanding problem for future research.

In order to constrain the time of the craton-lithosphere destruction, we focussed on cratons with diamondiferous kimberlites. Root loss, at some point after the craton formation, can also be inferred in other areas, where Archean cratonic crust occurs but is not underlain, at present, by thick mantle lithosphere (Fig. 5), for example, eastern Kalahari and north-western Congo Cratons (adjacent to the Karoo and Cameroon Line basalts, respectively).

In a recent paper, Hu et al.[13] also considered possible effects of plumes on the lithosphere of Africa. Using inferences from older, smooth tomographic models[74] and from the evolution of topography, they proposed that the depleted, buoyant lower part of the cratonic lithosphere beneath parts of Africa's cratons was removed by plumes but then replaced, fairly rapidly, by a new, fertile, dense lower lithosphere, so that the lithosphere remained thick but changed in its composition. A broader inference from this hypothesis is that cratonic roots are episodically removed or thinned by mantle dynamics but then re-grow, regaining their characteristic thickness.

The new evidence from our high-resolution tomography, based on much more data than previous tomography of Africa, confirms the erosion and pin-points its locations. It also shows, however, that the thinning of the lithosphere by plumes is permanent and irreversible, in contrast to the hypothesis of Hu et al.[13].

A further inference from our results is that the weakened, thin lithosphere beneath cratons affected by the lithospheric erosion is likely to be vulnerable to reworking in the next orogenic cycle[20]. The parts of cratons with eroded lithosphere thus have reduced chances of survival for a geologically long time.

The presence of hot, positively buoyant asthenosphere can be expected to increase the surface elevation[13]. Seismic velocities we observe in the asthenosphere and transition zone beneath southern Africa (Fig. 1d) are lower than elsewhere beneath the continent, indicating higher temperature, which can account for at least some of the higher elevation of southern Africa at present.

We conclude that Africa has lost a substantial proportion of its cratonic lithosphere over the last 200 My. During this time, Africa was moving slowly across an area with numerous plumes[75], which appear to have eroded its cratons, resulting in a more complex and fragmented cratonic lithosphere distribution than in Eurasia, North America or Australia. More generally, this implies that the total volume of cratonic lithosphere globally must have decreased substantially since its Archean formation, with the fate of each craton depending on its plate-tectonic movements and its luck in dodging mantle plumes.

## Methods

**Waveform tomography.** Our azimuthally anisotropic, S-wave speed, tomographic model is constrained by over 1.2 million seismograms, waveform-fitted using the Automated Multimode Inversion (AMI)[76] of surface- and S-waveforms. The global waveform dataset is from recordings of 6360 seismic stations, using 27550 earthquakes in total (Supplementary Fig. 2). Our model is focussed on the Africa region, with the data coverage in this region maximised (all freely available broadband data were included) and the regularisation tuned to optimise the resolution in Africa. The global data complements the regional dataset and ensures dense sampling of the entire Africa and surroundings. Every model-grid node of the model is sampled by at least 7767 paths (Fig. 1b). Our model-grid nodes have the same coordinates at different depths, so that the number of paths hitting a node does not change with depth, but the structural sensitivity of the data varies from node to node in 3D, as can be seen in the variations of the sums of the columns of the sensitivity matrix (Supplementary Fig. 2). The earthquakes are taken from the catalogue of Centroid-moment tensors from the Global Centroid-moment-tensor (GCMT) project[77], since 1994 and since 1990 for selected stations. Source-receiver distances are between 500 and 18,000 km.

After preprocessing (quality control, response correction) our waveform inversion procedure comprises three steps. First, we invert the seismogram waveforms using the well-established AMI[76]. AMI computes synthetic

seismograms by mode summation and performs waveform fitting of S-, multiple S- and fundamental-mode surface waves to the real data within multiple time-frequency windows, with elaborate time-frequency and phase weighting. The result is a set of linear equations with uncorrelated uncertainties[78] describing average, depth-dependent S- and P-wave velocity perturbations within 3D sensitivity volumes between the source and receiver with respect to a 3D reference model[76,79]. In the second step, the equations are inverted together as a large linear system for the 3D distribution of P- and S-wave velocities and S-wave azimuthal anisotropy[78,79], using LSQR[80]. The model is parametrised using a triangular grid with an average 327 km inter-knot spacing and with a depth parametrisation over 18 and 10 triangular basis functions for S- and P-wave velocities, respectively (S-wave velocities: 7, 20, 36, 56, 80, 110, 150, 200, 260, 330, 410−, 410+, 485, 585, 660−, 660+, 809 and 1007 km; P-wave velocities: 7, 20, 36, 60, 90, 150, 240, 350, 485 and 585 km). We invert for S-wave azimuthal anisotropy to ensure anisotropy does not map into isotropic heterogeneities but leave its interpretation for future work. Our 3D reference model comprises CRUST2[81] for the crust, with added topography and bathymetry, and our own 1D average for the upper mantle[79]. In the third step of our inversion procedure, we exploit the data redundancy to select only the most mutually consistent data by means of a posteriori outlier analysis. From an initial 3D model, we compute the synthetic data by matrix multiplication of the model and the sensitivity matrix. We then compare the synthetic and real data and discard the data with the largest misfits. This amounts to the selection of the most mutually consistent data and is effective at removing the data with the largest errors, in particular in the source location and origin time. In this study, we selected the most mutually consistent ∼770 thousand seismogram waveform fits to constrain our final model. Our linear 3D inversion is regularised by means of Laplacian lateral smoothing, vertical gradient damping and slight norm damping[79]. Depth-dependent regularisation parameters were tuned using synthetic spike tests with S-wave velocity anomalies at each depth node at selected locations on the model (Supplementary Figs. 7 and 8), to make sure that the maximum of the broadened output anomaly was at exactly the depth of the input spike, for each depth node. Supplementary Figs. 7 and 8 show example spike tests using the final regularisation parameters and scaling factors. We chose these examples to represent both best and worst case scenarios in our model. Vertical $\delta V_S$ profiles across the anomalies show coincident maxima for the input and retrieved synthetic models. Supplementary Fig. 9 shows an example of a targeted resolution test (its designed detailed in Supplementary Note 2), confirming, in particular, that if the lithosphere of Africa's cratons was not fragmented, as we observe, then the fragmentation would not appear in the model as an artifact.

**Kimberlites and lamproites database**. We used a global compilation of 4244 kimberlite sample locations[18]. The database includes kimberlites and lamproites that are diamond bearing, barren and with unknown-diamond content. In our analysis, we used African samples that were either diamond-bearing or with unknown-diamond content, aiming to include samples for which the database gives no information on the diamond content but which are from well-known diamondiferous regions (e.g. Angola, Tanzania). To avoid the sampling bias generated by pipes containing many samples, we re-sampled the data on a 0.03-degree grid, with samples in the same cell counting as one datum (Supplementary Note 3, Supplementary Fig. 10). We complemented the spatial information from this database with age information on confirmed diamondiferous kimberlite occurrences from[65].

## Data availability
The data supporting our findings are available from the corresponding author upon reasonable request.

## Code availability
The codes used to compute the tomographic model and all derived results are available from the corresponding author upon reasonable request.

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

## Acknowledgements

We thank Maya Kopylova, Balz Kamber, Dave Chew and Emma Tomlinson for discussions on the origin and composition of kimberlites and of cratons. We acknowledge the Incorporated Research Institutions for Seismology (IRIS; http://www.iris.edu), the GEOFON Global Seismic Network (https://geofon.gfz-potsdam.de) and Observatories and Research Facilities for European Seismology (http://www.orfeus-eu.org) for providing the data used in this study. We acknowledge Africa Array[28] for creating important parts of the dataset used in this study. The maps and cross-sections were implemented with the Generic Mapping Tools (GMT, http://gmt.soest.hawaii.edu), and ParaView (https://www.paraview.org). This work was supported by the Science Foundation Ireland (SFI) grant 13/CDA/2192, with additional support from grant 16/IA/4598, co-funded by SFI, the Geological Survey of Ireland, and the Marine Institute and from the SFI grant 13/RC/2092, co-funded under the European Regional Development Fund. We also acknowledge support from the Research Council of Norway, through its Centre of Excellence scheme, project number 223272 (CEED).

## Author contributions

N.C. and S.L. wrote the paper, with help from the other authors. N.C. performed tomographic inversions, with assistance from A.S. and S.L. C.G. provided the plate-tectonic reconstructions and contributed to discussions regarding kimberlite data distribution in the Angola and Kaapvaal regions. All authors jointly contributed to discussions.

## Competing interests

The authors declare no competing interests.
