## [Peer Review File (with redactions) · Nature Communications]

Editorial Note: Parts of this Peer Review File have been redacted as indicated where we could not obtain third party permissions

Reviewers' comments:

Reviewer #1 (Remarks to the Author):

Review of "African cratonic lithosphere carved by mantle plumes" by Celli, Lebedev, Schaeffer and Gaina

This paper presents a high-resolution tomography model in Africa using a large, newly available dataset. The model reveals a fragmented cratonic lithosphere of Africa at present, which is not observed in earlier tomography models. The authors compare the tomography at present to diamondiferous kimberlite distribution in the past to make a case that widespread lithospheric erosion took place over the last 200 Ma. The topic of this paper is of interest to the geology community, although the idea is not novel considering Hu et al. (2018) has just published similar ideas. The tomography itself is an important augmentation to the data available in Africa. The main story presented is likely to be true. However, I have some concerns about the arguments made by the authors. Some of the details presented remain questionable after scrutiny. Therefore, I recommend major revision before publication.

1. The authors assume that diamond-bearing kimberlites and lamproites exist only in cratons. This is generally true, but I am not sure if exceptions exist or not. In supplementary Fig. 5c, I do see many kimberlite sites that have a relatively low seismic velocity (blueish and orangish). Do they indicate locations where craton was eroded?

2. As indicated by this paper, some earlier studies (e.g. McKenzie and Priestley, 2007; Faure et al., 2011) suggested that kimberlites tend to erupt near the edges of cratonic mantle-lithosphere blocks. And the authors have used supplementary Fig. 7 to address this hypothesis. However, the close-to-boundary region is defined to be within 150 km or 200 km. I am wondering how the authors get this number. Outside of Africa, are the kimberlites not associated with craton erosion all located either within the defined boundary region or inside the cratons? In eastern Australia, there are many kimberlite samples, but the region is non-cratonic.

I notice some kimberlites are located within the mobile belts, also mentioned by (Yaxley et al., 2013, Nat. Commun). How to explain these kimberlite samples?

3. The emplacement ages of the kimberlite samples seem important for this paper, because following the authors' arguments, the latest age would represent the upper bounds of the time when the craton gets eroded. Therefore, the ages may give us clues to the causes of craton erosion.

The authors may refer to Tappe et al., 2018, EPSL, for age information. The data set there may not cover the ages of all the kimberlite samples in this paper. But it is still helpful to show the latest available age in a certain region.

4. Following No. 3, Fig. 10 in Tappe et al., 2018, EPSL, showed there were kimberlite emplacement after 60 Ma in southernmost Africa. But the hot spots overlapped this region at an age > 110 Ma as shown in Fig. 5. How to explain the age discrepancy?

5. I am not sure why the authors choose >5% positive velocity anomaly to outline the cratonic lithosphere. The change of this number will apparently affect the size of cratons and thus the fragmentation of cratons. For example, choosing >2% as craton would make the cratons appear more intact. Most non-cratonic (geologically defined) regions show neutral to negative velocity anomaly. This seems to suggest the authors could have chosen a lower number to characterize the cratons. In addition, it is intuitive that the edge of cratons may show lower seismic velocity as they may be warmer.

6. Tomography remains the only evidence for the fragmentation of cratonic lithosphere at present. If it is true, are there any other evidence to support it? Has the fragmentation caused topography change? Whether it has induced volcanisms or seismicity if it is still working? It may be good to think about these questions.

7. Not all lithosphere heterogeneity can be explained by plumes. For example, the model shows the Western African craton is divided into two pieces (supplementary Fig. 4). There are not hot spots in between. What could be the cause?

In addition, I don't see apparent topography change within the Western African craton. The whole craton seems a geologically quiet place. This impels me to think whether the tomography could resolve these features given the sparsity of data in this region even though new data were included.

8. There were kimberlite emplacement in TZ craton in Quaternary (Dawson, 1994). This challenges the authors' idea about the missing of TZ craton at depths if kimberlite does indicate thick craton.

Other comments:

1. The synthetic structure test does not serve the purpose of the paper well. The key of this model is the fragmentation of cratons. I would suggest to design a test where two craton blocks are placed close to each other to see where the model could recover the gap between the two cratons, say the gap between the two pieces in West Africa or the gap between IB and Congo.

2. Line 87-88. There are a lot of fast anomalies beneath the Atlantic Ocean at depths between 260- 485 km (supplementary Fig. 3). If the plume curved the craton or the continental breakup at > 100 Ma, which caused lower lithosphere detachment, they should now be present at deep mantle. Therefore, it is consistent with the notion proposed by Begg et al. (2009) and Hu et al. (2018).

3. Fig. 1b. Not sure how "node" is defined? Because I imagine the nodes are distributed in 3D, but here only a 2D map is shown. Do the authors sum up all the hits on the nodes vertically?

4. Fig. 3. Please add in the color bars for topography and seismic velocity.

5. Fig. 4. Please add horizontal scale and directions to the figure.

6. Supplementary Fig. 10. Delete one "simulating".

References:

Dawson, J. B. "Quaternary kimberlitic volcanism on the Tanzania Craton." *Contributions to Mineralogy and Petrology* 116.4 (1994): 473-485.

Yaxley, Gregory M., et al. "The discovery of kimberlites in Antarctica extends the vast Gondwanan Cretaceous province." *Nature communications* 4 (2013): 2921.

Tappe, Sebastian, et al. "Geodynamics of kimberlites on a cooling Earth: Clues to plate tectonic evolution and deep volatile cycles." *Earth and Planetary Science Letters* 484 (2018): 1-14.

Reviewer #2 (Remarks to the Author):

The paper of N.L. Celli et al. "African cratonic lithosphere carved by mantle plumes" deals with a topic of broad interest to the solid Earth community carrying out research on the ancient cores of continents and their evolution over the last 200 Ma.

This manuscript procures very good insights into the relationship between the lithospheric thickness and diamondiferous kimberlite fields providing important information on cratonic lithosphere destruction. The paper is very well written and illustrated. Given the investigated subject, it is of potential wide interest. I only have some comments and questions that may help clarify the work presented.

My principal concern is that almost all main results of presented here tomographic model AF2018 – such as 1) no extension of the high-velocities anomalies into the Atlantic Ocean; 2) two major high-velocity bodies beneath the Man-Léo and Reguibat shields; 3) absence of any cratonic mantle

beneath the Angolan shield; 4) the hidden Niassa craton; 5) thick cratonic lithosphere underneath the Arabian platform etc. – seems to be, at first sight, inferable from previous tomographic models as well (e.g. from the model SL2013 (Schaeffer & Lebedev, *Geophysical Journal International*, 2013), see Suppl. Fig. 4). The authors thus should provide more detail, comparing their tomographic model with previous ones. The presented Suppl. Figure 4 do not contain enough information to evaluate the claims made by the text (lines 49-59, 83-94). The addition of data such as S-wave velocity anomalies at different depths (for example) for different tomographic models and a corresponding discussion would allow the reader to judge the advantages of new model much more clearly.

Other points:

Lines 15-16, 284-285. Is it possible to provide a quantitative estimation of the craton-lithosphere loss?

Line 26. Reference to (Guillou-Frottier et al., *Global and Planetary Change*, 2012) might be also suitable here.

Lines 40-42. The relationship between the cratons mentioned here and Archean shields shown on Fig.1 is not clear. The Kalahari craton is not indicated on Fig. 1.

Lines 42-48. That's also not clear how these "Large Igneous Provinces" are related to "volcanic areas" shown on Fig.1. Corresponding discussion should be provided.

Line 75. I guess that "Fig. 1" should be "Fig. 1c-d".

Lines 91-92. The authors claim to "resolve sharply defined boundaries" for the cratons. However, the contours of the Congo and Kalahari cratons (Suppl. Fig. 7) appear to be very different for different threshold values of dVs.

Lines 97-98. The relation between LAB and "bottom of these cores" should be specified more precisely. Indeed, it is quite strange that the authors describe the results of the tomography in terms of "thicker" or "thinner" lithosphere without providing the map of the lithospheric thickness (in km) itself.

Lines 104-106. This statement might be appropriate for the introduction section to clarify the relations between the cratons and Archean shields (see comment above).

Line 117. None of the threshold values of dVs on Suppl. Fig. 7 do permit to distinguish "three distinct cores" mentioned here.

Lines 129-131. The Bangweulu block has been shown to be an essential element of the lithospheric structure of the central EARS. Together with the neighboring Tanzanian craton, it splits the vertical flow of a mantle plume into three sections that could trigger the development of the Eastern branch, the Western branch and the Malawi rift (Koptev et al., *Terra Nova*, 2018).

Lines 135-140. The authors claim to map, "in the first time", the Niassa craton. This isolate area of high-velocity anomalies, however, seems to be clearly distinguished on the previous seismic tomography model SL2013 as well (see Suppl. Fig. 4f).

Lines 178-179. Additional map showing the age distribution of the kimberlite intrusions may be useful for the discussion and interpretation of the results.

Line 231. Both geochemical (George et al., *Geology*, 1998; Rogers et al., *EPSL*, 2000; Nelson et al., *Geochim. Cosmochim. Acta*, 2008) and geophysical (e.g. Chang & Van der Lee, *EPSL*, 2011) evidences suggest two distinct mantle plumes – Kenyan plume and Afar plume – beneath East Africa. I guess the authors meant the Kenyan plume here.

Lines 230-236. The effect of mechanical erosion by plumes on cratonic roots is shown to be very limited (e.g. Wang et al., *Gcubed*, 2015; Koptev et al., *Nature Geoscience*, 2015). Melt infiltration may produce additional weakening ultimately leading to craton destruction but anhydrous decompression melting below cratons is insufficient even in the presence of a mantle plume (Zheng et al., *EPSL*, 2015). The authors therefore should provide additional discussion on the possible mechanisms of the Tanzanian craton destruction (e.g. say more on the rheological weakening induced by hydrous fluids and the possible sources of them).

Supplementary information. Suppl. Fig. 10 has not been referenced in the main text.

Alexander Koptev

We would like to thank the reviewers for their constructive comments and suggestions, which we have followed in revising our manuscript.

We detail our revisions and responses to all of the reviewers' inquiries below, point-by-point. The comments from the reviewers are reproduced in their entirety in italic. Additions and substantial changes to the text are referred to using the line numbers as in the revised manuscript with highlighted changes that we are submitting.

Reviewers' comments:

Reviewer #1 (Remarks to the Author):

Review of "African cratonic lithosphere carved by mantle plumes" by Celli, Lebedev, Schaeffer and Gaina

This paper presents a high-resolution tomography model in Africa using a large, newly available dataset. The model reveals a fragmented cratonic lithosphere of Africa at present, which is not observed in earlier tomography models. The authors compare the tomography at present to diamondiferous kimberlite distribution in the past to make a case that widespread lithospheric erosion took place over the last 200 Ma. The topic of this paper is of interest to the geology community, although the idea is not novel considering Hu et al. (2018) has just published similar ideas. The tomography itself is an important augmentation to the data available in Africa. The main story presented is likely to be true. However, I have some concerns about the arguments made by the authors. Some of the details presented remain questionable after scrutiny. Therefore, I recommend major revision before publication.

Thank you for the suggestions. Regarding the novelty, our observations, results and ideas are very different from those by Hu et al. Hu et al. proposed, based on inferences from older, lower-resolution tomography and from the evolution of topography, that the depleted, buoyant lower part of the cratonic lithosphere was removed by plumes and then replaced, fairly rapidly, by fertile, dense lower lithosphere, so that the lithosphere remained thick but changed in its composition. Their broader inference is that cratonic roots are episodically removed by mantle dynamics but then re-grown.

In contrast to this view, our results show a thinning of the lithosphere by plumes that is permanent and irreversible. The new lower lithosphere hypothesized by Hu et al. is absent, as our new tomographic model shows. We emphasize that only the very smooth, older-generation tomographic models show very broad high-velocity anomalies extending beneath the entire major cratons, which is due to these models' lower resolution, caused by a lot less data being available 10-20 years ago and by the a priori smoothness of those models. Newer models, including our group's SL2013 (Schaeffer and Lebedev 2013), SEMUM2 (French and Romanowicz 2015) or 3D2016_09Sv (Debayle et al. 2016), plotted in our Supplementary Fig. 4, all show greater fragmentation of the lithosphere, with cratonic lithosphere absent, for example, beneath the Angola Shield, which Hu et al. used as an example location with lithospheric "regrowth," based on the over-smoothed, older-generation tomography. Our new model now resolves the lithospheric complexity and fragmentation with greater detail yet, sufficient for its relationship with the distribution of kimberlites to emerge clearly.

The weakened, thin lithosphere that remains after the erosion is likely to be vulnerable to reworking in the next orogenic cycle, so that this part of the craton is unlikely to survive for a geologically long time. A fundamental direct inference from our observations is, then, that the total volume of the cratonic lithosphere has been decreasing with time due to plumes, in contrast to Hu et al.'s inferences, which implied that the lateral extent and thickness of cratons are preserved but their lithosphere may change in its composition. Our conclusions also do not require the puzzling scenario (proposed by Hu et al.) in which cold, fertile mantle material forms the new lower lithosphere of

cratons over ~100 m.y. or less and manages to remain stable rather than drip into the mantle below, despite its negative buoyancy.

In the first version of the manuscript, we opted not to debate Hu et al. explicitly. We now correct this and discuss this clearly and in detail in the revised manuscript (lines 322-353).

1. The authors assume that diamond-bearing kimberlites and lamproites exist only in cratons. This is generally true, but I am not sure if exceptions exist or not. In supplementary Fig. 5c, I do see many kimberlite sites that have a relatively low seismic velocity (blueish and orangish). Do they indicate locations where craton was eroded?

Yes, the P-T conditions required for the diamond stability field require the presence of very thick lithospheric roots, characteristic of cratons. We had stated this on lines 205-209 and referred to the classic papers that have established this. We have now expanded the explanation (lines 209-217).

Because thick cratonic lithosphere was present when the diamondiferous kimberlites erupted, it is not surprising when the thick lithosphere is still present in these locations today (see the histogram in the Supplementary Fig. 5c). The locations where it is not present include the well-documented areas where cratonic lithosphere has been eroded, including the Wyoming and North China Cratons (Supplementary Fig. 5c), mentioned explicitly on lines 239-241. They also include the areas of cratonic erosion in Africa that we identify in this study. These areas, which we focus on in particular (parts of the Kalahari and West African Cratons and the Angola Shield), are recognized cratons, with Archean crust observed at the surface.

We would consider it likely that many of the remaining exceptions (in South America or on Greenland's western coast, for example) are also where the cratonic lithosphere has been eroded. Making this point, however, would require specific discussion of the geology and evolution of these regions, which is beyond the scope of this paper.

Finally, some of the exceptions may indeed include diamondiferous kimberlites that do not come from cratonic lithosphere but, perhaps, from as deep as the mantle transition zone (410-660 km depth range), as suggested by Ringwood et al. (1992). This might be the case along the eastern coast of Australia, where kimberlites are found in Phanerozoic orogens. A very small minority, these would, however, be the kind of an exception that confirms the rule.

We now cite Ringwood et al. (1992) and mention the possibility of unusually deep origin of some Kimberlites (line 212). In Africa, however, thermo-barometry studies summarized in Begg et al. (2009) and other papers cited in the manuscript yield abundant evidence for the origin of diamondiferous kimberlites within the cratonic lithosphere.

2. As indicated by this paper, some earlier studies (e.g. McKenzie and Priestley, 2007; Faure et al., 2011) suggested that kimberlites tend to erupt near the edges of cratonic mantle-lithosphere blocks. And the authors have used supplementary Fig. 7 to address this hypothesis. However, the close-to-boundary region is defined to be within 150 km or 200 km. I am wondering how the authors get this number.

The different boundary widths tested were, in fact, 200 and 300 km wide (200 km and +/-150 km from the boundary line), and the boundary lines were defined using a number of alternative dVs thresholds. This accounted for the finite resolution of our tomography and for different definitions

one may choose to define the craton boundary. Our inference (no preferential location of kimberlites within craton-boundary areas) was robust, supported by every test, regardless of how the boundary was defined. Making the boundaries much broader would result in most or all of the relatively small cratons considered getting included into a boundary area, making the comparison meaningless.

We have now phrased the definitions of the boundary areas clearer than in the original version of the paper in the Supplementary material (lines 15-23) and in the Fig. S7 caption.

Outside of Africa, are the kimberlites not associated with craton erosion all located either within the defined boundary region or inside the cratons? In eastern Australia, there are many kimberlite samples, but the region is non-cratonic. I notice some kimberlites are located within the mobile belts, also mentioned by (Yaxley et al., 2013, Nat. Commun). How to explain these kimberlite samples?

Most kimberlites globally are, indeed, located within cratons (e.g., our Supplementary Fig. 5c). We do not, however, extend our entire analysis to the global scale and do not attempt to define cratonic boundaries globally. Our primary evidence here is the new tomographic model of Africa, and our primary focus---the cratons of Africa. Quantitative analysis of cratonic boundaries elsewhere---although, of course, interesting---would be well beyond the scope of this paper.

The samples along the eastern coast of Australia appear to be the most anomalous on the entire global map. As discussed above, they are a small, highly anomalous minority and as such can be seen as an exception that confirms the rule---certainly a potential target for important future studies. Elsewhere, cratonic lithosphere can, indeed, underlie crustal units that have been reworked sufficiently to be identified as mobile belts. We discuss such hidden cratons (Niassa, eastern Arabia) in the text.

The discovery of the first known kimberlites in Antarctica (Yaxley et al., 2013) is exciting, but they are not diamondiferous, as far as we know, and therefore do not necessarily indicate cratonic lithosphere beneath their location.

3. The emplacement ages of the kimberlite samples seem important for this paper, because following the authors' arguments, the latest age would represent the upper bounds of the time when the craton gets eroded. Therefore, the ages may give us clues to the causes of craton erosion. The authors may refer to Tappe et al., 2018, EPSL, for age information. The data set there may not cover the ages of all the kimberlite samples in this paper. But it is still helpful to show the latest available age in a certain region.

We agree and we have used the dataset by Tappe et al. 2018, to which we refer at lines 241, 251, 271, 275, 281, 293 and 421. We have now also added the diamondiferous-kimberlite age ranges from Tappe et al. 2018 to Fig. 3a, for clarity.

4. Following No. 3, Fig. 10 in Tappe et al., 2018, EPSL, showed there were kimberlite emplacement after 60 Ma in southernmost Africa. But the hot spots overlapped this region at an age > 110 Ma as shown in Fig. 5. How to explain the age discrepancy?

This figure includes non-diamondiferous (barren) kimberlites, which do not present evidence for the occurrence of thick cratonic lithosphere. We explain in the paper (lines 205-217) that it is, specifically, the diamondiferous kimberlites that are proxies for the presence of thick cratonic lithosphere at the time of their eruption.

5. I am not sure why the authors choose >5% positive velocity anomaly to outline the cratonic lithosphere. The change of this number will apparently affect the size of cratons and thus the fragmentation of cratons. For example, choosing >2% as craton would make the cratons appear more intact. Most non-cratonic (geologically defined) regions show neutral to negative velocity anomaly. This seems to suggest the authors could have chosen a lower number to characterize the cratons. In addition, it is intuitive that the edge of cratons may show lower seismic velocity as they may be warmer.

Large (>5%) shear-wave velocity (V_s) anomalies beneath cratons are well known to seismologists and come out clearly in global statistical analyses (e.g., Schaeffer and Lebedev 2015). Also, as we now note clearly on lines 122-123, our results and inferences are not dependent on this particular number and also hold with a 4% or 4.5% threshold (Supplementary Fig. 7).

Lowering the threshold to as low as 2%, however, would broaden the range of lithosphere included well beyond the range characteristic of cratons. The much higher V_s anomalies within cratonic lithosphere are evidenced, apart from tomography, by samples from the mantle. At a 150 km depth beneath a moderately thick craton (Fig. 8 in McKenzie et al. 2005), temperatures of 1000-1060 C have been estimated, based on P-T data from mantle samples (McKenzie et al., 2005). In thicker cratonic lithospheres, including those beneath Congo and West Africa Cratons (Fig. S3 shows pronounced high-velocity anomalies at least down to 260 km depth), temperature at this depth is likely to be even lower. Converting V_s to temperature using computational petrology (Fig. 8 in Agius & Lebedev 2005), we infer that V_s at 150 km within cratonic lithosphere must be around 4.6 km/s or higher. Computing average V_s profiles across Africa for different anomaly percentages (Fig. L1, this letter), we see that V_s anomaly in the 2-3% range gives V_s values much lower than that. (A 2% anomaly w.r.t. the global average, suggested by the Reviewer 1, will yield values very close to the continental average values, not similar to the anomalous cratonic values.) V_s values exceeding 4.6 km/s, consistent with the mantle sample data, are found for anomalies in the 4-5% range or above, consistent with the thresholds we use in the paper. The choice of a particular threshold is not critical, and we had already made sure to plot cratonic boundaries using different values (4, 4.5, 5% - Figs. 1, S7); the boundaries of the cratons with these values are similar and support our conclusions in each case. A 2% threshold suggested by Reviewer 1, by contrast, would be clearly inconsistent with what we know about cratonic lithosphere from seismology and from the samples from kimberlites (Fig. 8 in McKenzie et al. 2005).

We now explain this briefly in the text (lines 122-123), also citing the papers that we reproduced the figures from (Fig. 8 in McKenzie et al. 2005, Fig. 8 in Agius & Lebedev 2005)

6. Tomography remains the only evidence for the fragmentation of cratonic lithosphere at present. If it is true, are there any other evidence to support it? Has the fragmentation caused topography change? Whether it has induced volcanisms or seismicity if it is still working? It may be good to think about these questions.

Good questions. Regarding topography, we have added the following text on lines 344-353:

The relationship of the lithospheric thickness and topography in cratons depends on the composition of the lithosphere (Fullea et al. 2012, Ravenna et al. 2018). For example, replacing a depleted, neutrally buoyant lithosphere with neutrally buoyant asthenosphere will have no effect on topography (Fullea et al. 2012). Also, the large effect on topography of relatively small variations in the bulk density of the crust can obscure the lithospheric thickness-topography relationship (Ravenna et al. 2018). The presence of hot, positively buoyant asthenosphere, however, can be expected to increase the surface elevation (Hu et al. 2018). Seismic velocities we observe in the asthenosphere and transition zone beneath southern Africa (Fig. 1d) are lower than elsewhere beneath the continent, indicating higher temperature, which can account for at least some of the higher elevation of southern Africa at present.

Regarding volcanism, Large Igneous Province volcanism and kimberlites provide key evidence on the lithospheric evolution and lithospheric structure in the past, which we use throughout the paper. Seismic imaging, however, provides the most abundant evidence on the lithospheric structure at present.

Our tomography provides an advance in resolution at the scale of the entire Africa. Our results can be compared, at a few locations in Africa, with those of smaller-scale, regional studies using arrays of seismic and magnetotelluric stations. The thinning of the Tanzania Craton and the presence of low-velocity asthenosphere beneath it has been observed by Weeraratne et al. (2003), for example, which we point out on lines 147-148. Pronounced heterogeneity of the lithospheric structure to the south of Tanzania, consistent with our model, has been detected by seismic (O'Donnell et al. 2013) and MT (Sarafian et al. 2017) regional imaging, which we cite on lines 159-160.

Regarding lithospheric deformation (giving rise to seismicity), we have now added a paragraph on the role of the Niassa Craton in localizing the deformation associated with the southward propagation of the EARS at the Malawi Rift, along the boundary of the craton (lines 162-169).

7. Not all lithosphere heterogeneity can be explained by plumes. For example, the model shows the Western African craton is divided into two pieces (supplementary Fig. 4). There are not hot spots in between. What could be the cause?

That is certainly true: not all cratonic lithosphere heterogeneity is due to plumes. Variations in the lithospheric thickness are observed in cratons around the world and, in particular, in Africa, as evidenced by our model and by regional studies independent from ours, a few of which are cited in the paper (O'Donnell et al. 2013; Sarafian et al. 2017; Ravenna et al. 2018). The configuration of the thick-lithosphere blocks within the West African Craton may have been preserved since the Precambrian assembly of this craton. Alternatively, the lithospheric thickness within it may have changed due to mantle dynamic processes. We do not have evidence for these, however, apart from the kimberlites in the western part of the craton that we discuss in the paper.

In addition, I don't see apparent topography change within the Western African craton. The whole craton seems a geologically quiet place. This impels me to think whether the tomography could resolve these features given the sparsity of data in this region even though new data were included.

Our resolution tests show that if the west African craton was a single, broad feature seen in the older, smooth models (Begg et al. 2009; Ritsema et al. 2011), we would retrieve it as such

(Supplementary Fig. 10). We point this out in the Resolution tests section of the Supplementary material and now also on lines 411-414.

The parts of the West African Craton without the very thick cratonic lithosphere do, nevertheless, have moderately thick lithosphere (high-velocity anomalies of 2-3%, Fig. 1) and are, indeed, stable, tectonically quiet regions. This is in contrast to the thin, warm lithosphere and probably hot asthenosphere to the east, where the low velocities shown by our model coincide with high topography and hotspot volcanism.

8. There were kimberlite emplacement in TZ craton in Quarternary (Dawson, 1994). This challenges the authors' idea about the missing of TZ craton at depths if kimberlite does indicate thick craton.

The Paleogene kimberlites in Tanzania (pre-plume) are diamondiferous. By contrast, the more recent ones (following the plume impact), which the reviewer is referring to, are non-diamondiferous, consistent with the erosion of the lithosphere and the loss of the P-T conditions necessary for the diamond formation. We explain this on lines 269-273. The thin lithosphere here is evidenced both by our tomography and by the regional study of Weeraratne et al. (2003) (lines 147-148).

Other comments:

1. The synthetic structure test does not serve the purpose of the paper well. The key of this model is the fragmentation of cratons. I would suggest to design a test where two craton blocks are placed close to each other to see where the model could recover the gas between the two cratons, say the gap between the two pieces in West Africa or the gap between IB and Congo.

We agree that the resolution testing is important. Our tests are designed to target the main question regarding the seismic tomography evidence used in the paper: is there or is there not fragmentation of the lithosphere? Here, we essentially follow the reviewer's line of inquiry in the second part of their comment 7. If we used fragmented lithosphere in an input model and got fragmented lithosphere in the output, this would not answer the question whether or not the fragmentation may be an artifact. If, instead, we use smooth, non-fragmented lithosphere in the input model and resolve it accurately in the output, with no artificial fragmentation appearing, this confirms that the fragmentation is not an artifact of uneven coverage. We show and discuss this in Supplementary Fig. 10, in the Resolution tests section of the Supplementary material, and now also on lines 411-414.

2. Line 87-88. There are a lot of fast anomalies beneath the Atlantic Ocean at depths between 260-485 km (supplementary Fig. 3). If the plume curved the craton or the continental breakup at > 100 Ma, which caused lower lithosphere detachment, they should now be present at deep mantle. Therefore, it is consistent with the notion proposed by Begg et al. (2009) and Hu et al. (2018).

Yes, there are many fast anomalies with an amplitude around 2% in the deep upper mantle beneath the South Atlantic. These may indeed indicate portions of continental lithosphere removed by convection.

What we meant, however, was different: some of the older tomographic models (e.g., Begg et al. 2009) seemed to show strong anomalies at lithospheric depths (over 5% at 50-175 km depths) extending from the Congo Craton westward beneath the Atlantic Ocean. This was probably an artifact, as confirmed by our model and other recent models (our SFig. 4), but this is still used, sometimes, in geodynamic arguments. It is important to point out that the currently intact cratonic lithosphere is contained within the coastlines of Africa. We now re-phrase this clearer on lines 92-99.

3. Fig. 1b. Not sure how “node” is defined? Because I imagine the nodes are distributed in 3D, but here only a 2D map is shown. Do the authors sum up all the hits on the nodes vertically?

We now explain this clearer on lines 371-374:

Our model-grid nodes have the same coordinates at different depths, so that the number of paths hitting a node does not change with depth, but the structural sensitivity of the data varies from node to node in 3D, as can be seen in the variations of the sums of the columns of the sensitivity matrix (Supplementary Fig. 1).

4. Fig. 3. Please add in the color bars for topography and seismic velocity.

Done, thanks.

5. Fig. 4. Please add horizontal scale and directions to the figure.

Done, thanks.

6. Supplementary Fig. 10. Delete one “simulating”.

Done, thanks.

Reviewer #2 (Remarks to the Author):

The paper of N.L. Celli et al. "African cratonic lithosphere carved by mantle plumes" deals with a topic of broad interest to the solid Earth community carrying out research on the ancient cores of continents and their evolution over the last 200 Ma.

This manuscript procures very good insights into the relationship between the lithospheric thickness and diamondiferous kimberlite fields providing important information on cratonic lithosphere destruction. The paper is very well written and illustrated. Given the investigated subject, it is of potential wide interest. I only have some comments and questions that may help clarify the work presented.

My principal concern is that almost all main results of presented here tomographic model AF2018 – such as 1) no extension of the high-velocities anomalies into the Atlantic Ocean; 2) two major high-velocity bodies beneath the Man-Léo and Reguibat shields; 3) absence of any cratonic mantle beneath the Angolan shield; 4) the hidden Niassa craton; 5) thick cratonic lithosphere underneath the Arabian platform etc. – seems to be, at first sight, inferable from previous tomographic models as well (e.g. from the model SL2013 (Schaeffer & Lebedev, Geophysical Journal International, 2013), see Suppl. Fig. 4). The authors thus should provide more detail, comparing their tomographic model with previous ones. The presented Suppl. Figure 4 do not contain enough information to evaluate the claims made by the text (lines 49-59, 83-94). The addition of data such as S-wave velocity anomalies at different depths (for example) for different tomographic models and a corresponding discussion would allow the reader to judge the advantages of new model much more clearly.

We accept the criticism---this was unclear, and we have now clarified this in the text. Our arguments in the original version of the manuscript were developed with other models in mind (in particular, the earlier ones from Begg et al. (2009) or Ritsema et al. (2011), lacking regional detail but used very

widely in the geological literature). Our group's own SL2013 has, indeed, already shown more detail than such older, smoother models. The key point for the present study, however, is that the newly increased resolution brings regional-scale lithospheric structure into sharper focus, sufficiently for the kimberlite-craton relationship we examine to emerge clearly. We would not be able to do our quantitative analysis with earlier models, including SL2013. Looking at southern Africa, for example, the Kalahari Craton in SL2013 is generally where we see it now but it is smoother and broader; the western boundary of the West African Craton is also less clearly defined.

We have revised the corresponding parts of the text and now explain more clearly the advances of our tomographic model relatively to previously published ones (lines 69-73 and 99-106). In order to facilitate a more comprehensive model comparison, we have also added more map views at different depths and velocity gradient information for the models compared in Supplementary Fig. 4.

Lines 15-16, 284-285. Is it possible to provide a quantitative estimation of the craton-lithosphere loss?

The histogram in Fig. 3 suggests that at least around 50% of cratonic lithosphere in Africa sampled by known kimberlites has been lost. We now modify the sentence on line 219 to say that at least half of the kimberlites are not atop cratonic lithosphere today. If kimberlites sampled cratonic lithosphere uniformly, this would yield an estimate of the total proportion of cratonic lithosphere that has been eroded. Unfortunately, the sampling by kimberlites is, instead, highly uneven. For this reason, it would be difficult to estimate the proportion of lost lithosphere accurately. Our focus is, instead, on establishing the occurrence of the erosion of cratonic lithosphere, which, as we show, took place beneath a substantial proportion of cratons of Africa.

Line 26. Reference to (Guillou-Frottier et al., Global and Planetary Change, 2012) might be also suitable here.

Done, thanks.

Lines 40-42. The relationship between the cratons mentioned here and Archean shields shown on Fig.1 is not clear. The Kalahari craton is not indicated on Fig. 1.

Done, thanks.

Lines 42-48. That's also not clear how these "Large Igneous Provinces" are related to "volcanic areas" shown on Fig.1. Corresponding discussion should be provided.

The caption for Figure 1 has been clarified, and the definition of Large Igneous Provinces added in lines 42-43 to clarify the difference between LIPs and volcanics, thanks.

Line 75. I guess that "Fig. 1" should be "Fig. 1c-d".

Done, thanks

Lines 91-92. The authors claim to "resolve sharply defined boundaries" for the cratons. However, the contours of the Congo and Kalahari cratons (Suppl. Fig. 7) appear to be very different for different threshold values of dVs.

In Supplementary Fig. 7 we show a wide range of possible craton definitions, using dVs thresholds down to 3.5%, which is very low. Our purpose here is to demonstrate that kimberlites do not plot preferentially near craton boundaries even when the boundary is defined so as to include the periphery of cratons with relatively thin lithosphere. With 4-5% dVs threshold, the inferred boundaries of cratons vary only a reasonably small amount. The variations in the location of the boundaries can be considered as indicative of uncertainty in the definition of the boundary. The 5% threshold used in the main text yields the most conservative boundary definition, with the periphery of cratons where the lithosphere gets thinner not included.

Lines 97-98. The relation between LAB and “bottom of these cores” should be specified more precisely. Indeed, it is quite strange that the authors describe the results of the tomography in terms of “thicker” or “thinner” lithosphere without providing the map of the lithospheric thickness (in km) itself.

We now explain this in some detail on lines 115-123, as follows:

In Fig. 2, we plot the highest-velocity (and, by inference, lowest-temperature) cores of the cratonic mantle lithosphere using 3D surfaces of positive 5% Vs anomaly. This threshold isolates velocities characteristic of cratonic lithosphere (according to global tomography (Schaeffer 2015), temperature estimates from samples from cratonic mantle lithosphere (McKenzie et al. 2005) and conversion of the temperatures to seismic velocities (Agius and Lebedev, 2013)). The bottom of these cores is not the Lithosphere-Asthenosphere Boundary (LAB). However, thicker (and colder) cores do indicate where the lithosphere is the thickest (Supplementary Fig. 2), as expected from the relationship between the lithospheric thickness and temperature given by realistic geotherms (Eeken et al. 2018; Garber et al. 2018). In the thick cratonic lithosphere, the increase of temperature with depth is relatively slow and the LAB is expected to be marked by only a subtle change in the slope of the depth dependence of temperature and seismic velocity (Ravenna et al. 2018). For this reason, direct estimates of the LAB depth from seismic tomography models are ambiguous, unless thermodynamic modelling including seismic data or models is performed (Fullea et al. 2012). For the purpose of discriminating whether or not the characteristically cold, thick cratonic lithosphere is present beneath a location, the 5% Vs anomaly is an effective threshold. Our results and inferences, however, are not dependent on this particular number and also hold with a 4% or 4.5% threshold (Supplementary Fig. 7).

Lines 104-106. This statement might be appropriate for the introduction section to clarify the relations between the cratons and Archean shields (see comment above).

Yes, there would be advantages to that, but this statement leads to the ones after it, so that including it in the introduction would imply including these following sentences as well. We opted to keep the introduction brief and expand on this shortly below, in the section “Cratonic lithosphere of Africa.”

Line 117. None of the threshold values of dVs on Suppl. Fig. 7 do permit to distinguish “three distinct cores” mentioned here.

The cores are distinct in the sense that they are units with particularly thick lithosphere separated by units with thinner lithosphere. This thinner lithosphere is still cratonic, but the occurrence of the thick-lithosphere cores is still interesting and worth pointing out.

On lines 141-142, we now add a new sentence to make this clear: “Between these blocks, the lithosphere is also cratonic, but thinner than within them.”

Lines 129-131. The Bangweulu block has been shown to be an essential element of the lithospheric structure of the central EARS. Together with the neighboring Tanzanian craton, it splits the vertical flow of a mantle plume into three sections that could trigger the development of the Eastern branch, the Western branch and the Malawi rift (Koptev et al., Terra Nova, 2018)

The suggested publication prompts important questions regarding the development of the rifts. The Bangweulu block, according to our model, has an intermediate-thickness lithosphere. The strength of this lithosphere certainly has a role in the localization of the deformation associated with the rifting. Looking at the entire rift system, the Eastern and Western Branches have developed along the perimeter of the Tanzania Craton (a mechanically strong unit). The Malawi Rift to the south has developed along the eastern side of the Niassa Craton that we map in this study (another strong unit). We now spell out this important implication of our findings on lines 162-169, also citing the Koptev et al. (2018) paper.

Lines 135-140. The authors claim to map, “in the first time”, the Niassa craton. This isolate area of high-velocity anomalies, however, seems to be clearly distinguished on the previous seismic tomography model SL2013 as well (see Suppl. Fig. 4f).

SL2013 displays a small, high velocity anomaly centred a few hundred km southwest from the centre of the Niassa Craton. The location and lateral extent of this anomaly are substantially different, so that it cannot be considered the same feature. SL2013 did not have the data sampling to resolve the Niassa Craton accurately.

Lines 178-179. Additional map showing the age distribution of the kimberlite intrusions may be useful for the discussion and interpretation of the results.

Age information added in Fig. 3, thanks.

Line 231. Both geochemical (George et al., Geology, 1998; Rogers et al., EPSL, 2000; Nelson et al., Geochim. Cosmochim. Acta, 2008) and geophysical (e.g. Chang & Van der Lee, EPSL, 2011) evidences suggest two distinct mantle plumes – Kenyan plume and Afar plume – beneath East Africa. I guess the authors meant the Kenyan plume here.

Corrected to “Kenyan Plume” (line 268), reference to George et al. (1998) inserted.

Lines 230-236. The effect of mechanical erosion by plumes on cratonic roots is shown to be very limited (e.g. Wang et al., Gcubed, 2015; Koptev et al., Nature Geoscience, 2015). Melt infiltration may produce additional weakening ultimately leading to craton destruction but anhydrous decompression melting below cratons is insufficient even in the presence of a mantle plume (Zheng et al., EPSL, 2015). The authors therefore should provide additional discussion on the possible mechanisms of the Tanzanian craton destruction (e.g. say more on the rheological weakening induced by hydrous fluids and the possible sources of them).

This point adds to the discussion, and we have added a mention of the possible enhancement of pre-existing metasomatic weakening on lines 306-307. We have also added the citation of the Koptev et al. (2015) paper, relevant here.

More generally, we hope that our study poses useful, important questions for future investigations of the mechanisms of the weakening of cratonic lithosphere by mantle plumes, leading to its destruction. In our paper, we cite published geochemical and geodynamic studies that focused on the mechanisms of craton destruction by thermo-chemical plumes (Sobolev et al. 2011; Wang et al. 2015). New original geochemical data analysis or numerical geodynamic modelling are beyond the scope of this study. At the same time, a speculative, non-quantitative discussion would be something that, we feel, could weaken rather than strengthen the paper. Instead, we refer the reader to the geodynamics- and geochemistry-focussed papers on the subject by Sobolev et al. (2011) and Wang et al. (2015).

Supplementary information. Suppl. Fig. 10 has not been referenced in the main text.

Alexander Koptev

Reference to SFig 10 has been added on lines 411-414.

[redacted]

Fig. 8 in McKenzie et al. 2005

[redacted]

Figure L1. Vs profiles that average over all Vs columns across Africa with Vs anomaly (averaged over the 80-150 km depth interval) in the specified percentage range.

References

Afonso, J.C., Fernandez, M., Ranalli, G., Griffin, W.L. and Connolly, J.A.D., 2008. Integrated geophysical-petrological modeling of the lithosphere and sublithospheric upper mantle: Methodology and applications. *Geochemistry, Geophysics, Geosystems*, 9(5).

Agius, M.R. and Lebedev, S., 2013. Tibetan and Indian lithospheres in the upper mantle beneath Tibet: Evidence from broadband surface-wave dispersion. *Geochemistry, Geophysics, Geosystems*, 14(10), pp.4260-4281.

Begg, G. C. et al. 2009. The lithospheric architecture of Africa: Seismic tomography, mantle petrology, and tectonic evolution. *Geosphere* 5, pp.23–50.

Debayle, E., Dubuffet, F. & Durand, S. An automatically updated S -wave model of the upper mantle and the depth extent of azimuthal anisotropy. *Geophys. Res. Lett.* **43**, 674–682 (2016).

French, S. W. & Romanowicz, B. Broad plumes rooted at the base of the Earth's mantle beneath major hotspots. *Nature* **525**, 95–99 (2015).

Fullea, J., Afonso, J.C., Connolly, J.A.D., Fernandez, M., García-Castellanos, D. and Zeyen, H., 2009. LitMod3D: An interactive 3-D software to model the thermal, compositional, density, seismological, and rheological structure of the lithosphere and sublithospheric upper mantle. *Geochemistry, Geophysics, Geosystems*, 10(8).

Fullea, J., Lebedev, S., Agius, M.R., Jones, A.G. and Afonso, J.C., 2012. Lithospheric structure in the Baikal–central Mongolia region from integrated geophysical-petrological inversion of surface-wave data and topographic elevation. *Geochemistry, Geophysics, Geosystems*, 13(8).

Hu, J. et al. Modification of the Western Gondwana craton by plume–lithosphere interaction. *Nat. Geosci.* (2018). doi:10.1038/s41561-018-0064-1

McKenzie, D., Jackson, J. and Priestley, K., 2005. Thermal structure of oceanic and continental lithosphere. *Earth and Planetary Science Letters*, 233(3-4), pp.337-349.

O'Donnell, J. P., Adams, A., Nyblade, A. A., Mulibo, G. D. & Tugume, F. The uppermost mantle shear wave velocity structure of eastern Africa from Rayleigh wave tomography: Constraints on rift evolution. *Geophys. J. Int.* **194**, 961–978 (2013).

Ravenna, M., Lebedev, S., Fullea, J. & Adam, J. M. C. Shear-Wave Velocity Structure of Southern Africa's Lithosphere: Variations in the Thickness and Composition of Cratons and Their Effect on Topography. *Geochemistry, Geophys. Geosystems* 1–20 (2018). doi:10.1029/2017GC007399

Ringwood, A. E., Kesson, S. E., Hibberson, W. & Ware, N. Origin of kimberlites and related magmas. *Earth and Planetary Science Letters* 113, 521–538 (1992).

Ritsema, J., Deuss, A., Van Heijst, H. J. & Woodhouse, J. H. S40RTS: A degree-40 shear-velocity model for the mantle from new Rayleigh wave dispersion, teleseismic traveltimes and normal-mode splitting function measurements. *Geophys. J. Int.* **184**, 1223–1236 (2011).

Sarafian, E. *et al.* Imaging Precambrian lithospheric structure in Zambia using electromagnetic methods. *Gondwana Res.* **54**, 38–49 (2017).

Schaeffer, A.J. and Lebedev, S., 2013. Global shear speed structure of the upper mantle and transition zone. *Geophysical Journal International*, 194(1), pp.417-449.

Schaeffer, A.J. and Lebedev, S., 2015. Global heterogeneity of the lithosphere and underlying mantle: a seismological appraisal based on multimode surface-wave dispersion analysis, shear-velocity tomography, and tectonic regionalization. In *The Earth's Heterogeneous Mantle* (pp. 3-46). Springer.

Sobolev, S. V. *et al.* Linking mantle plumes, large igneous provinces and environmental catastrophes. *Nature* **477**, 312–316 (2011).

Tappe, S., Smart, K., Torsvik, T., Massuyeau, M. & de Wit, M. Geodynamics of kimberlites on a cooling Earth: Clues to plate tectonic evolution and deep volatile cycles. *Earth Planet. Sci. Lett.* **484**, 1–14 (2018).

Weeraratne, D. S., Forsyth, D. W., Fischer, K. M. & Nyblade, A. A. Evidence for an upper mantle plume beneath the Tanzanian craton from Rayleigh wave tomography. *J. Geophys. Res.* **108**, 2427 (2003).

Wang, H., Van Hunen, J. & Pearson, D. G. The thinning of subcontinental lithosphere: The roles of plume impact and metasomatic weakening. *Geochemistry Geophys. Geosystems* **18**, 1541–1576 (2015).

Yaxley, G. M. *et al.* The discovery of kimberlites in Antarctica extends the vast Gondwanan Cretaceous province. *Nat. Commun.* **4**, 1–7 (2013).

REVIEWERS' COMMENTS:

Reviewer #1 (Remarks to the Author):

The authors provide an improved manuscript based on reviewer comments. Thus, I did not have much more comment to do on the manuscript and it can be published in the present form.

Reviewer #2 (Remarks to the Author):

The authors have appropriately answered my comments and suggestions, and revised the paper with new experiments in line with my suggestions. From my point of view it would deserve to be published.

Juho Rousu

Reviewer #3 (Remarks to the Author):

The authors have addressed all my concerns. I believe the dataset presented in the manuscript will add great value to the community.

Reviewers' comments:

Reviewer #1 (Remarks to the Author):

I am satisfied with the authors' response to my comments related to kimberlite. Their arguments about diamondiferous kimberlite helps clarify many things, including the age of craton deconstruction and the different implications of diamondiferous vs. barren kimberlite on the thickness of cratons.

Thank you for this, and for the further comments.

Here are a few remarks about the interpretation of their tomography models.

1. SEMUM2 (French and Romanowicz 2015) is one of the recent tomography, but the cratons it has resolved is definitely not fragmented as the authors have claimed in the rebuttal letter. I see an intact southern Kaapvaal Craton. Note the fragmentation of this craton is a key feature for the authors' arguments about craton evolution.

In fact, most of the models in Supplementary Fig. 4 (including SL2013, 3D2016_09Sv, SEMUM2 and the older ones) at 100 km show a Kaapvaal Craton extending further south than the model in this study AF2019.

The older, smoother models, examples of which are shown in SFig. 4 (S40RTS and CUB, in the two columns on the right), show smooth high-velocity anomalies extending beneath the entire 3 major cratons (West Africa, Congo and Kalahari, the latter including Kaapvaal as its southern part). The more recent models all show substantial heterogeneity within the boundaries of these three major cratons, including the absence of cratonic lithosphere in parts of them. This is the fragmentation we are referring to.

Furthermore, all the recent models agree that there is no thick cratonic lithosphere beneath the Angola Shield, for example, or that the portion of the Kalahari craton with a thick lithosphere is much smaller than seen in the previous-generation, smoother models. When we look at even smaller scales, the recent model also show differences, of course. We believe our model provides a substantial advance in resolution. However, our inferences and conclusions are not based or critically dependent on any single feature (for example, on the exact location of the southern boundary of Kaapvaal Craton's thick cratonic lithosphere). Our inferences are required by many features taken together, some of them (such as the absence of the cratonic lithosphere beneath the Angola Shield or a greatly reduced lateral extent of the cratonic lithosphere beneath Kalahari) are consensus features among recent models and others are not. The fragmentation itself is now emerging as a consensus feature among recent models.

Reviewer 2 wondered earlier if our model may be too similar to its predecessor for us to talk of new discoveries. Reviewer 1 now wonders if our model may be suspiciously different from previous ones. Both are valid questions, and we welcome the scrutiny—these are the sort of questions we did ask ourselves when constructing the new model. The truth is that the new, higher-resolution model shares its larger-scale (and some of its smaller scale) features with the earlier models. But its higher resolution also brings into focus smaller-scale features than could not be resolved previously, and this now enables the patterns we focus on to emerge clearly.

Let's take the difference as the advance of AF2019 made through methods or data in the recent couple of years. But still, in AF2019, I see fast anomalies beneath southern Kaapvaal below 150 km and extending at least to 200 km (Supplementary Fig. 3). If they are not lithosphere whether regrown or original, what are they? Some artifacts?

Relatively weak Vs anomalies (under 3%) are present beneath most cratons (as well as other continental units) globally. Beneath some cratons, they are positive and beneath others—negative. This is part of the global heterogeneity of the upper mantle that is ubiquitous, according to a large majority of models (although not entirely understood at present). These anomalies are unlikely to all be artifacts.

Specifically, we agree that the relatively weak positive Vs anomaly at 150-200 km beneath southern Kaapvaal Craton may well indicate upper-mantle material cooled from above. We note, however, that the Vs (and, by inference, thermal) anomaly here is much lower in amplitude than within intact cratonic lithosphere.

2. I don't think >5% Vs as the threshold for cratonic lithosphere is agreed among earth scientists or seismologist.

Using the material provided by the authors in the rebuttal letter, the temperature of cratonic lithosphere at 150 km is between 1000 C – 1060 C. The ambient mantle is about 1300 C. This gives the temperature anomaly 240 C – 300 C. From Fig. 8 in Agius & Lebedev 2005, the reference Vs at 150 km is about 4.4 km/s. Temperature anomaly of -200 C has Vs around 4.51 km/s. Temperature anomaly of -400 C has Vs around 4.61 km/s. Let's assume the cratonic lithosphere at 150 km has a temperature anomaly of -300 C, which gives the value around 4.56. Now do the math, I get Vs anomaly = (4.56 – 4.4)/4.4 = 3.6%. From Supplementary Fig. 7, changing the threshold from 5% to 3.5% has already changed the appearance of cratons a lot.

First of all, we agree with the reviewer that our inferences and conclusions should not depend critically on an arbitrarily picked number. We believe they do not, and we believe the paper shows clearly that they do not. The average dVs in Africa at the locations of the diamondiferous and “unknown content” kimberlites is 3.7%, rising to 4.1% for the diamondiferous ones only (Fig. 3). This is much lower than the 5.2% and 5.7%, respectively, observed globally (Fig. S5c). Our study certainly does not aim to say the last word on the exact southern boundary of the cratonic lithosphere of the Kaapvaal Craton, for example. However, the pattern of a large proportion of African kimberlites being located on relatively warm and thin lithosphere, compared to the global distribution, is clear and robust. (As the map in Fig. 3 shows, numerous kimberlites are where dVs is lower than 3.0% as well.)

Regarding the specific numbers, Figure Fig. 8 in Agius & Lebedev 2005, gives the absolute values of temperature and Vs, so that we can avoid subtractions and additions and potential round-off uncertainties. (We reproduce below the three figures as they were in the previous letter.) If we look at the geotherms at the 150 km depth (the panel on the right), the temperature slightly over 1000C is given by the orange line (the third line from the right). Looking at Vs (the panel on the left), this corresponds to Vs values exceeding 4.6 km/s (the third line from the left). According to Fig L1, values greater than 4.6 km/s correspond to Vs in the 4-5% or 5-6% range, which is why we picked 5%. We also tested lower thresholds, as we point out in the text (lines 121-123: “Our results and inferences, however, are not dependent on this particular number and also hold with a 4\% or 4.5\% threshold.”)

We would like to emphasize once more that possible uncertainties in the estimates above would not change our main inferences and conclusions, which are based on a clear and robust pattern (a large

proportion of the diamondiferous kimberlites in Africa are located on relatively warm and thin lithosphere, in stark contrast to the global distribution – see Fig. 3 for Africa and Fig. S5c for global).

In tomography models, there must be smearing around the edge of craton which could either enhance or further lower the resultant Vs anomaly, as evidenced by the synthetic tests (Supplementary Fig. 10).

Yes, there is lateral averaging along a characteristic averaging length (resolution length). This does not affect, however, our main observations (a large proportion of the diamondiferous kimberlites in Africa are located on relatively warm and thin lithosphere, in contrast to the global distribution), which stands with different dVs thresholds. The changes in the threshold values result in inferred craton boundaries moving by distances comparable to the possible effect the lateral averaging at the actual boundary.

In addition, if the cratonic lithosphere regrew after deconstruction, they are likely not as cool as the original lithosphere (thermal cooling needs time). The authors have no ways to exclude the possibility that the fast shear anomaly around 2-3(or 2-4?)% near the cratonic boundary or below 150 km are likely the regrown lithosphere. Indeed, if the tomography model is accurate, those beneath the southern Kaapvaal are very likely to be the regrown lithosphere, especially those extending beyond 150 km which should be warmer according to the geotherm of craton and thus require even lower value of Vs for a regrown lithosphere.

We agree: if a lithosphere undergoes substantial thinning, it will subsequently tend to cool and grow in thickness. It is important to also note, however, that the global distribution of very thick lithospheres exclusively beneath cratons indicates that Phanerozoic lithospheres are unlikely to reach the thickness and thermal anomaly comparable to those in cratons, probably due to their lack of the compositional buoyancy (created, beneath the Archean cratons, in the course of the high-degree melting facilitated by the hotter Archean geotherms).

As for the Agnolan shield, I agree with the authors that the center part of the shield seems permanently removed. But Hu et al. 2019 was suggesting a broader removal of cratonic lithosphere. There are signals that the northern and southern part of the shield may have regrown, as evidenced by the 2-3% fast anomalies. Therefore, it is likely both the authors' model and the model of Hu et al. 2019 are working in this region.

Yes, absolutely. We very much agree that the lithosphere will increase in thickness subsequently to a catastrophic thinning. The inferences of the two studies are, indeed, in agreement in this regard.

3. L332-338: those sentences are misleading. Hu et al. 2019 suggested the delaminated cratonic lithosphere can regrow, because of the sharp thermal gradient at the interface where the lithosphere is removed. It is basic physics that the mantle below the interface will be cooled down, which causes growth of the cratonic lithosphere. There must be some degree of growth. The question is whether it can restore the initial cratonic geotherm.

We agree with the argument. There will be some degree of growth, no doubt. The question is, indeed, whether or not the original cratonic geotherm can be restored. Our model shows that there is, of course, some continental mantle lithosphere present beneath the locations of kimberlites. But

in many locations, it is not as thick and cold as required for the occurrence of diamonds, which indicates that it has been eroded and not regrown to the original thickness. This is consistent with the general expectation—confirmed by the global correlation of the occurrence of very thick lithosphere and Archean crust (e.g., Schaeffer and Lebedev 2015)—that compositional buoyancy, acquired in the course of the Archean partial melting, is required for the lithosphere to grow to the characteristic cratonic thickness (>200 km). This was our argument in the text.

This argument, however, is not essential for our paper. As suggested by the reviewer, we now remove the sentences on the lines 341-347 (previously 332-338), inserted in the previous revision.

The authors suggest negative, but their argument is misleading or incomplete. First, Phanerozoic lithosphere is not as thick as the craton, maybe simply because Phanerozoic lithosphere is much younger (<500 Ma) and it usually undergoes deformation. In contrast, the Archean craton is much more stable (higher viscosity as it is more depleted and cooler, the remaining 150 km thick cratonic lithosphere in Hu's model is likely strong enough to resist deformation due to horizontal stresses) and much older (>2.5 Ga, cools to greater depth). Second, whether the regrown lithosphere would delaminate or not depends both on density anomaly and viscosity. Assuming a thermal expansion of $3e-5 \text{ K}^{-1}$, temperature anomaly of 200-300 C could only contribute to density anomaly of 0.6%-0.9%. In comparison, eclogite has a density anomaly of several percent. The fact that eclogitized lower crust at subduction zones does not always founder is because of the strong viscosity. An increase of temperature from ambient mantle temperature by 200-300 C could result in several magnitudes of increase in viscosity, due to strong temperature dependence of viscosity.

I suggest the authors remove these sentences. The argument is misleading and not well supported.

As suggested by the reviewer, we now remove the sentences on lines 341-347 (previously 332-338).

4. L343-352: misleading again. Hu et al. suggested a compositionally denser lower cratonic lithosphere. Replacing this layer with neutrally buoyant asthenosphere causes some degree of uplift, which they use to explain the present topography.

In fact, the present topography is probably not the key to understand the whole process, because there are large uncertainties. The change of the topography in the past is equally important. The authors seem to suggest the cratonic lithosphere is neutrally buoyant. If this is true, we would not observe late Cretaceous uplift in the region as suggested in Hu et al.

I suggest the authors remove these sentences too, because these are unfair comments, some even represent misunderstanding. I was meant that the authors might look into the history of topography evolution to better understand the whole process. This is likely out of scope.

We agree that this debate may take us well beyond the scope of the study. As suggested by the reviewer, we now remove most of the sentences on lines 352-361 (previously 343-352), apart from the last two that are not in disagreement with the reviewer's comment.

Overall, I agree with the publication of this manuscript, given the amount of the work and the fact that it is of broad interest and can inspire a lot of discussion. In the meantime, the authors are discussing tiny features for a large-scale tomography model. They need to be very careful about their interpretation. They also need to persuade why the readers should trust more on their model rather

than other earlier models (I don't see a strong reason yet). In addition, I don't think anyone should have the confidence to make a strong argument about the permanent removal of lithosphere, especially in south Kaapvaal. The authors need to tone it down.

Agreed: the lithosphere is not removed, although it has been thinned. We have removed the two passages that raised the reviewers' objections.

[redacted]

Figure L1. Vs profiles that average over all Vs columns across Africa with Vs anomaly (averaged over the 80-150 km depth interval) in the specified percentage range.

References

- Afonso, J.C., Fernandez, M., Ranalli, G., Griffin, W.L. and Connolly, J.A.D., 2008. Integrated geophysical-petrological modeling of the lithosphere and sublithospheric upper mantle: Methodology and applications. *Geochemistry, Geophysics, Geosystems*, 9(5).
- Agius, M.R. and Lebedev, S., 2013. Tibetan and Indian lithospheres in the upper mantle beneath Tibet: Evidence from broadband surface-wave dispersion. *Geochemistry, Geophysics, Geosystems*, 14(10), pp.4260-4281.
- Fullea, J., Afonso, J.C., Connolly, J.A.D., Fernandez, M., García-Castellanos, D. and Zeyen, H., 2009. LitMod3D: An interactive 3-D software to model the thermal, compositional, density, seismological, and rheological structure of the lithosphere and sublithospheric upper mantle. *Geochemistry, Geophysics, Geosystems*, 10(8).
- Fullea, J., Lebedev, S., Agius, M.R., Jones, A.G. and Afonso, J.C., 2012. Lithospheric structure in the Baikal–central Mongolia region from integrated geophysical-petrological inversion of surface-wave data and topographic elevation. *Geochemistry, Geophysics, Geosystems*, 13(8).
- McKenzie, D., Jackson, J. and Priestley, K., 2005. Thermal structure of oceanic and continental lithosphere. *Earth and Planetary Science Letters*, 233(3-4), pp.337-349.
- Schaeffer, A.J. and Lebedev, S., 2013. Global shear speed structure of the upper mantle and transition zone. *Geophysical Journal International*, 194(1), pp.417-449.
- Schaeffer, A.J. and Lebedev, S., 2015. Global heterogeneity of the lithosphere and underlying mantle: a seismological appraisal based on multimode surface-wave dispersion analysis, shear-velocity tomography, and tectonic regionalization. In *The Earth's Heterogeneous Mantle* (pp. 3-46). Springer.

REVIEWERS' COMMENTS:

Reviewer #1 (Remarks to the Author):

No further comments.

Reviewer's comments

Reviewer #1 (Remarks to the Author):

No further comments.

We thank Reviewer 1 for the previous comments which helped us improve the manuscript.